

# A large increase in the carbon inventory of the land biosphere since the Last Glacial Maximum: constraints from multi-proxy data

Aurich Jeltsch-Thömmes[1,2], Gianna Battaglia[1,2], Olivier Cartapanis[2,3], Samuel L. Jaccard[2,3], and Fortunat Joos[1,2]

[1]Climate and Environmental Physics, Physics Institute, University of Bern, Bern. Switzerland
[2]Oeschger Centre for Climate Change Research, University of Bern, Bern, Switzerland
[3]Institute of Geological Sciences, University of Bern, Bern, Switzerland

*Correspondence to:* Aurich Jeltsch-Thömmes (jeltsch@climate.unibe.ch)

**Abstract.** Atmospheric $CO_2$ increased by about 90 ppm across the transition from the Last Glacial Maximum (LGM) to the end of the preindustrial (PI) period. The contribution of changes in land carbon stocks to this increase remains uncertain. Estimates of the PI-LGM difference in land biosphere carbon inventory ($\Delta$land) range from -400 to +1,500 GtC, based on upscaling of scarce paleo soil carbon or pollen data. A perhaps more reliable approach infers $\Delta$land from reconstructions of

the stable carbon isotope ratio in the ocean and atmosphere assuming isotopic mass balance with recent studies yielding $\Delta$land values of about 300–400 GtC. Surprisingly, however, earlier studies considered a mass balance for the ocean–atmosphere–land biosphere system only. Thereby, these studies neglect carbon exchange with sediments, weathering-burial flux imbalances, and the influence of the deglacial reorganization on the isotopic budgets. We show this neglect to significantly bias low deglacial $\Delta$land in simulations using the Bern3D Earth System Model of Intermediate Complexity v.2.0s. We constrain $\Delta$land to $\sim$850

GtC (median estimate; 450 to 1250 GtC $1\sigma$ range) by using reconstructed changes in atmospheric $\delta^{13}$C, marine $\delta^{13}$C, deep Pacific carbonate ion concentration, and atmospheric $CO_2$ as observational targets in a Monte Carlo ensemble with half a million members. Sensitivities of the target variables to changes in individual deglacial carbon cycle processes are established from factorial simulations over the past 21,000 years with the Bern3D model. These are used in the Monte Carlo ensemble and provide forcing–response relationships for future model–model and model–data comparisons. Uncertainties in the estimate of

$\Delta$land remain considerable due to model and proxy data uncertainties. Yet, it is likely that $\Delta$land is larger than 450 GtC and highly unlikely that the carbon inventory in the land biosphere was larger for the LGM than during the recent preindustrial period.

## 1 Introduction

Atmospheric $CO_2$ varied between about 180 and 300 ppm over the past 800,000 years (Neftel et al., 1982; Siegenthaler et al.,
2005; Lüthi et al., 2008; Marcott et al., 2014; Bereiter et al., 2015). These $CO_2$ variations were tightly coupled to glacial–interglacial climate change (Petit et al., 1999; Siegenthaler et al., 2005) and amplified orbitally driven climate variations (Jansen et al., 2007). Despite their climatic importance, the exact mechanisms behind these past $CO_2$ variations remain enigmatic. A wide range of explanatory processes that are related to the marine and terrestrial carbon cycle as well as to exchange processes



with reactive ocean sediments, coral reefs, and the lithosphere has been proposed (e.g. Archer et al., 2000; Sigman and Boyle, 2000; Sigman et al., 2010; Fischer et al., 2010; Menviel et al., 2012; Ushie and Matsumoto, 2012; Jaccard et al., 2013; Galbraith and Jaccard, 2015; Heinze et al., 2016; Wallmann et al., 2015). While the first simulations covering glacial–interglacial cycles, including dynamic ocean and land models, that represent reconstructed atmospheric $CO_2$ variability emerge, these models are

not yet able to reproduce variations in important proxy data and to represent the timing of $CO_2$ changes over the last glacial termination adequately (Brovkin et al., 2012; Menviel et al., 2012; Ganopolski and Brovkin, 2017). To make progress in this research area requires the combination of multiple lines of evidence where proxy reconstructions are compared to quantitative model analyses. This will help to constrain underlying processes and to quantify their contribution.

One of the processes is the storage of carbon in the land biosphere but its change over the last deglaciation is debated. Most

studies addressing past land biosphere carbon focus on the change in land biosphere carbon inventory between the Last Glacial Maximum around 21,000 years before present (LGM; 21 kyrBP) and the recent preindustrial period (PI). This inventory difference (PI minus LGM) is here termed Δland and includes changes in the various land biosphere reservoirs, such as plants, mineral soils, permafrost, peatland, yedoma, and wetlands as well as changes on shelves exposed during the LGM.

Available proxy-based Δland estimates (e.g. Shackleton, 1977; Adams et al., 1990; Bird et al., 1994; Adams and Faure,

1998; Zimov et al., 2006, 2009; Zech et al., 2011; Ciais et al., 2012; Peterson et al., 2014; Menviel et al., 2017) encompass values ranging from -400 GtC to 1,500 GtC. This is large compared to the current land biosphere stock amounting to around 4,000 GtC (Ciais et al., 2013) as well as to the deglacial atmospheric change of around 200 GtC (1 ppm is equivalent to 2.12 GtC). The high end of the Δland range, implying much smaller land carbon storage during the LGM than today, is based on estimates relying on pollen records (Adams et al., 1990; Van Campo et al., 1993; Crowley, 1995; Adams and Faure,

1998). Pollen and macrofossils recorded in various archives hold information on past vegetation distribution, but do not allow constraining carbon inventories neither in soils nor in vegetation. The low end is based on reconstructions of the carbon content in paleo soils (Zimov et al., 2006, 2009; Zech et al., 2011). These reconstructions are scarce, of local nature, and restricted to cold high-altitude or high-latitude regions, rendering extrapolations to the global scale speculative. In a recent data synthesis study, Lindgren et al. (2018) suggest that the loss of carbon from northern permafrost areas from the LGM to PI was

lower than initially suggested by Zimov et al. (2009) and that total land carbon storage in northern mid- and high-latitudes increased by about 400 GtC over the deglaciation. Simulations with spatially explicit terrestrial biosphere models generally yield a positive Δland (see references in Joos et al., 2004; Prentice et al., 2011; O'Ishi and Abe-Ouchi, 2013; Ganopolski and Brovkin, 2017; Davies-Barnard et al., 2017). These simulations are based on models, which typically do not simulate changes in neither wetland, peatland, and permafrost, nor carbon stocks in deep deposits of loess and yedoma, in lakes, and in subglacial

sediments (Tarnocai et al., 2009; Zimov et al., 2009; Yu et al., 2010; Dommain et al., 2014; Lindgren et al., 2018). Furthermore, the results are impaired by our limited understanding of how low atmospheric $CO_2$, generally colder and drier climate, altered nutrient fluxes, and largely exposed shelves in the tropics affected carbon inventories.

The classical (Shackleton, 1977) and arguably most reliable approach to reconstruct Δland considers the deglacial change in the marine stable carbon isotope signature as recorded in sediments and the mass balance of carbon and $^{13}C$. The argument is

that the uptake of isotopically light carbon by the land biosphere causes a corresponding, measurable perturbation in the average



$\delta^{13}$C value in the ocean–atmosphere system. The $^{13}$C mass balance approach, using marine sediment and ice core records, provided consistent and converging estimates of $\Delta$land with recent estimates ranging from 300 to 400 GtC (Curry et al., 1988; Duplessy et al., 1984; Crowley, 1991; Bird et al., 1994; Crowley, 1995; Bird et al., 1996; Joos et al., 2004; Ciais et al., 2012; Peterson et al., 2014; Menviel et al., 2017). Early estimates relied on spatially limited $\delta^{13}$C sediment records (Duplessy et al.,

1984; Curry et al., 1988; Bird et al., 1994, 1996) but recently, more comprehensive $\delta^{13}$C data compilations (Oliver et al., 2010; Peterson et al., 2014) became available and were also used for mean ocean $\delta^{13}$C value and $\Delta$land determinations applying a dynamic ocean model (Menviel et al., 2017). The details of the approach are outlined in the literature (e.g. Ciais et al., 2012).

All previous $^{13}$C mass balance studies assumed a closed system. They thus neglected the carbon and isotopic exchange between the ocean and reactive sediments, the input flux from rock weathering, and carbon burial in consolidated sediments by

focussing on the atmosphere–ocean–land system only. It has been argued (e.g. Ciais et al., 2012) that changes in weathering contributed little to the deglacial $CO_2$ rise and that large changes in the calcium carbonate ($CaCO_3$) burial in sediments (and coral reefs) are inconsistent with the reconstructed changes in the depth of the lysocline.

There are, however, several important reasons to question these arguments as well as the closed system assumption. The potential role of changes in $CaCO_3$ dissolution and burial for regulating past $CO_2$ has been widely discussed (e.g. Broecker

and Peng, 1987; Archer and Maier-Reimer, 1994; Archer et al., 2000; Cartapanis et al., 2018). Concerning the role of particulate organic carbon (POC), Tschumi et al. (2011) and Roth et al. (2014) identified a "nutrient–burial feedback" and demonstrated its importance also for $\delta^{13}$C for a range of mechanisms with the potential to explain low glacial $CO_2$. They conclude that the long-term balance between burial of organic material and tracer input into the ocean through weathering, typically neglected in earlier studies, should be considered when investigating the glacial–interglacial evolution of atmospheric $CO_2$, $\delta^{13}$C, and

related tracers.

The first specific reason to challenge the closed system assumption is the continuous weathering–burial cycle. The continuous flux of biogenic particles made out of calcium carbonate and organic matter carries any perturbation in $\delta^{13}$C in the coupled atmosphere–surface ocean system to ocean sediments (Tschumi et al., 2011; Menviel et al., 2012; Roth et al., 2014; Cartapanis et al., 2016). A second reason is the chemical buffering of land induced $CO_2$ perturbations by marine $CaCO_3$ sediments,

a process known as carbonate compensation (e.g. Archer et al., 2000; Broecker and Clark, 2001). Carbonate compensation causes a transfer of carbon and $^{13}$C between the ocean and reactive ocean sediments. Third, and of even greater importance to $\delta^{13}$C, are changes in the cycling and weathering-sedimentation imbalances of POC that have the potential to exert a large impact on the $^{13}$C budget as shown in earlier studies (Tschumi et al., 2011; Roth et al., 2014). These mechanisms should not be neglected, and a range of processes related to the deglacial reorganization of the marine carbon cycle likely influenced their

signal in the available proxy data (Schmitt et al., 2012; Yu et al., 2013; Peterson et al., 2014; Kerr et al., 2017; Qin et al., 2017; Luo et al., 2018) .

The documented reorganization of the marine carbon cycle (e.g. Sigman and Boyle, 2000; Sigman et al., 2010; Fischer et al., 2010; Elderfield et al., 2012; Ciais et al., 2013; Martinez-Garcia et al., 2014; Jaccard et al., 2016; Cartapanis et al., 2016) affected the transfer of $^{13}$C and carbon between the ocean–atmosphere system and reactive sediments and the lithosphere.

The reorganization includes changes in $CO_2$ solubility, ocean ventilation (e.g. Burke and Robinson, 2012; Sarnthein et al.,



2013; Skinner et al., 2017) and water mass distribution (e.g. Duplessy et al., 1988; Peterson et al., 2014; Lippold et al., 2016; Menviel et al., 2017; Gottschalk et al., 2018), air–sea gas transfer rates, for example via changes in sea-ice extent (e.g. Stephens and Keeling, 2000; Gersonde et al., 2005; Waelbroeck et al., 2009; Sun and Matsumoto, 2010), export (e.g. Kohfeld et al., 2005; Jaccard et al., 2013) and remineralization (e.g. Bendtsen et al., 2002; Matsumoto, 2007; Taucher et al., 2014) of

biogenic material associated with the cycling of organic carbon, $CaCO_3$, and biogenic opal (Tschumi et al., 2011; Roth et al., 2014; Matsumoto et al., 2014), changes in coral reef growth (e.g. Berger, 1982; Milliman and Droxler, 1996; Kleypas, 1997; Ridgwell et al., 2003; Vecsei and Berger, 2004; Menviel and Joos, 2012), and in the input of dust, nutrients and lithogenic material by atmospheric deposition (e.g. Lambert et al., 2015) and rivers, and coastal and shelve exchange processes (Ushie and Matsumoto, 2012; Wallmann et al., 2015). These changes influence the particle flux of organic and inorganic carbon,

opal, and mineral particles adding carbon and mass to the sediments. They further influence the partitioning between sediment dissolution and consolidation (burial) by altering the concentration of dissolved inorganic carbon (DIC) and $\delta^{13}C$ of DIC and POC, alkalinity, carbonate ion, oxygen, and other tracers in near-sediment bottom water and in sediment pore water.

The primary goal of this study is to provide a new, observationally constrained best estimate of $\Delta$land recognizing sediment interactions in an Earth system model. A second goal is to establish response sensitivities for a range of proxy targets to

changes in individual key deglacial carbon cycle mechanisms. We further scrutinize the closed system assumption in the $^{13}C$ mass balance approach. This is tackled by probing the role of the weathering–burial cycle and sediment compensation. We run idealized pulse-like carbon uptake simulations with the Bern3D Earth System Model of Intermediate Complexity (EMIC) both with and without simulating sediment interactions, and with and without interactive weathering. The results demonstrate the importance of these mechanisms and suggest that previous $\delta^{13}C$-based estimates of $\Delta$land have been systematically underes-

timated. Sensitivities of individual marine and terrestrial processes contributing to deglacial change are determined and varied in a large number of combinations to find all possible solutions that match a set of observational targets in a Bayesian Monte Carlo data assimilation framework (see Fig. 1).

## 2   Methods

### 2.1   The Bern3D model

The Bern3D EMIC is briefly described in the Appendix A, where also model spin-up and experimental settings for the impulse response experiments are detailed. The sediment module has been updated compared to previous work to include most recent observational information on sediment composition and fluxes (Cartapanis et al., 2016, 2018; Tréguer and De La Rocha, 2013) as detailed in Appendix B. Key model parameters were adjusted within their uncertainties to best reproduce the observation-based spatial distributions and global stocks of $CaCO_3$, POC, and opal within reactive sediments, observational estimates of

global burial and redissolution fluxes, and of euphotic-zone export fluxes of biogenic particles (Table 1). This updated version of the Bern3D model is introduced as Bern3D v2.0s. The simulated oceanic DIC inventory for PI is 37,175 GtC, similar to the 37,310 GtC estimated based on the GLODAP v.2 data set (Lauvset et al., 2016).





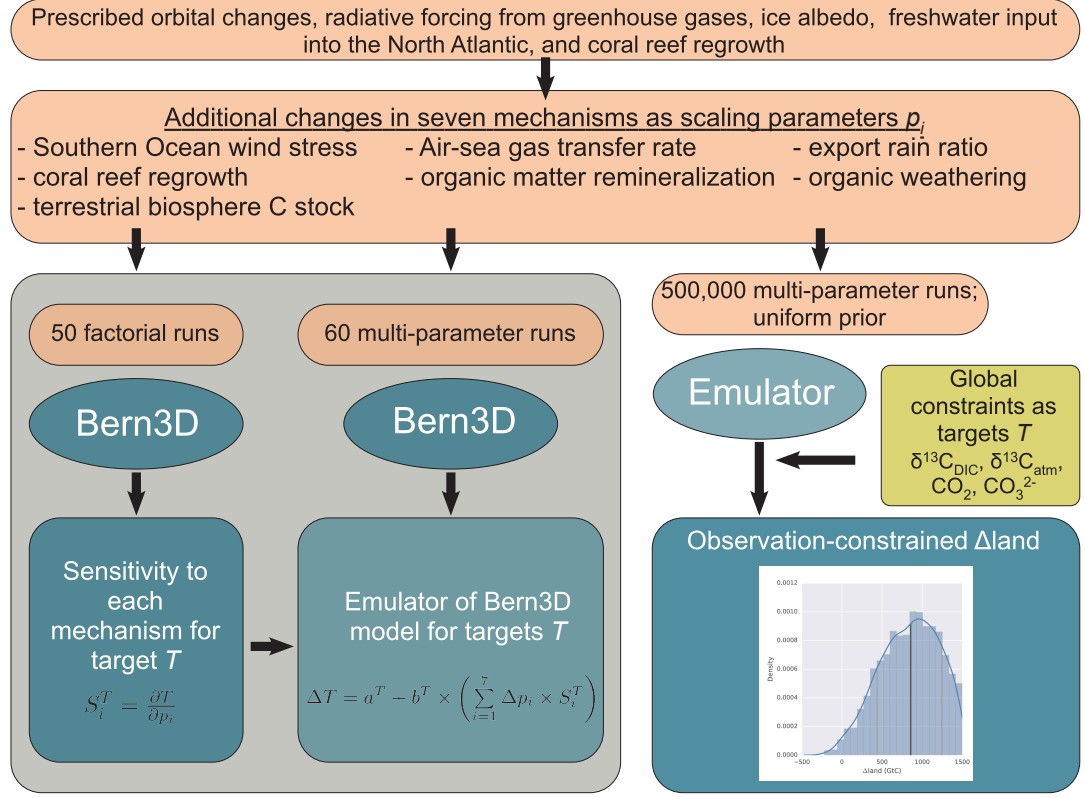

**Figure 1.** Flow chart outlining the steps applied to achieve an estimate of possible $\Delta$land values. To this end, seven generic deglacial carbon cycle mechanisms (Table 2 and Section 2.2) and four observational targets (Section 2.2.3) were included. The targets are the PI-LGM change in atmospheric $\delta^{13}$C ($\Delta\delta^{13}C_{atm}$) recorded in Antarctic ice (Schmitt et al., 2012); the PI-LGM change in the global ocean average $\delta^{13}$C of dissolved inorganic carbon ($\Delta\delta^{13}C_{DIC}$) as reconstructed from 480 benthic foraminiferal records (Peterson et al., 2014); the PI-LGM change carbonate ion concentration in the deep Pacific inferred from boron (B) to calcium (Ca) ratios in foraminifera (Yu et al., 2013); the PI-LGM change in atmospheric $CO_2$ (Petit et al., 1999; Monnin et al., 2001; Siegenthaler et al., 2005; Lüthi et al., 2008). The seven processes are varied individually by systematic parameter variations in addition to the well-established forcings (top of figure) in 50 factorial simulations with the Bern3D. This yields a first set of sensitivities or forcing–response curves for our four targets and seven processes. Next, we apply a Latin Hypercube parameter sampling to vary the processes in combination and probe for non-linear interactions in 60 multi-parameter simulations with the Bern3D model. The results are used to adjust the forcing response relationship; adjustments are small, except for $CO_2$ response curves (see Fig. C1 in the Appendix). Analytical representations of the forcing–response curves are used to build a simple emulator of the Bern3D model (Section 2.2.4). The emulator is applied in a Monte Carlo ensemble with half a million members to probe a wide range for each individual process. Finally, the ensemble results are constrained by the four constraints to yield a best estimate and an uncertainty distribution for $\Delta$land.





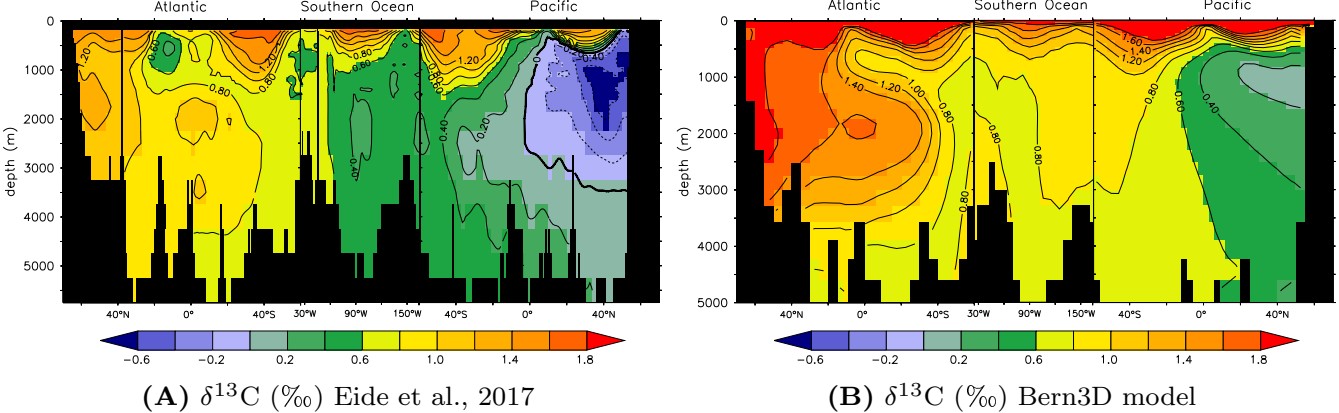

**(A)** $\delta^{13}$C (‰) Eide et al., 2017          **(B)** $\delta^{13}$C (‰) Bern3D model

**Figure 2.** Observed (A) versus simulated (B) pre-industrial $\delta^{13}C_{DIC}$ distributions along a cross section through the Atlantic (25°W), across the Southern Ocean (58°S) and into the Pacific (175°W). (A) shows data from Eide et al. (2017) based on ocean measurements and corrected for the Suess-effect. (B) shows $\delta^{13}$C as modeled in the Bern3D model under pre-industrial boundary conditions.

$^{13}$C stocks and exchanges are modeled in the atmosphere–ocean–sediment–landbiosphere system. $^{13}$C fluxes to the lithosphere associated with the burial of POC and CaCO$_3$ and weathering input to the ocean are explicitly simulated. The burial of POC and CaCO$_3$ results in an isotopic signature of the burial flux of around -9 ‰, intermediate between the isotopically light POC ($\sim$-20 ‰) and the isotopically heavy CaCO$_3$ signature ($\sim$2.9 ‰; see Appendix A for implementation of isotopic discrim-

ination). An isotopic perturbation in the atmosphere, e.g. through uptake of isotopically light carbon by the land biosphere, is transferred to the ocean through air–sea gas exchange and to the land by photosynthesis. Within the ocean, any perturbation signal is communicated between the surface ocean and the deep ocean by advection, convection, and mixing. In addition, POC and CaCO$_3$ export fluxes from the surface ocean communicate to ocean sediments and therefore to the burial flux.

Figure 2 shows modeled $\delta^{13}$C in a section through the Atlantic, Southern Ocean, and Pacific under pre-industrial (1765 CE)
boundary conditions with a prescribed atmospheric $\delta^{13}$C signature of -6.305 ‰. Compared to measured data corrected for the Suess-effect (Eide et al., 2017) it is visible that modeled $\delta^{13}$C values are biased high by a seemingly constant offset of about 0.4 ‰, while the $\delta^{13}$C patterns are captured by the model.

## 2.2   Transient LGM to PI sensitivity simulations

### 2.2.1   Standard transient forcings

Starting from PI steady state conditions, the model is first run for 20 kyr under constant LGM forcing and then for an additional 20 kyr under transient forcing from the LGM to PI. Forcings (Fig. 3) include radiative forcing imposed by CO$_2$, N$_2$O, and CH$_4$ (Joos and Spahni, 2008) and variations in orbital parameters (Berger, 1978). Radiative forcing of CO$_2$ is prescribed and biogeochemistry is calculated with interactive CO$_2$. Ice sheet extent and related changes in albedo are prescribed based on benthic $\delta^{18}$O data (Lisiecki and Raymo, 2005) and ice sheet reconstructions (Peltier, 1994). Additionally, freshwater (FW)





**Table 1.** Export, burial, and deposition fluxes as well as sediment inventories of $CaCO_3$, opal, and particulate organic carbon (POC) as determined in the model after a pre-industrial spin-up and literature estimates. References: [a] Battaglia et al. (2016), [b] Tréguer and De La Rocha (2013), [c] Sarmiento and Gruber (2006), [d] Milliman and Droxler (1996), [e] Feely et al. (2004)

| variable | units | Bern3D | observational estimates |
| --- | --- | --- | --- |
| $CaCO_3$ export | GtC yr$^{-1}$ | 0.98 | 0.72-1.05[a] |
| opal export | Tmol Si yr$^{-1}$ | 109.69 | 88-122[b] |
| POC export | GtC yr$^{-1}$ | 11.98 | 6.5-13.1[c] |
| $CaCO_3$ deposition | GtC yr$^{-1}$ | 0.49 | 0.5[d] |
| opal deposition | Tmol Si yr$^{-1}$ | 77.39 | 78.8[b] |
| POC deposition | GtC yr$^{-1}$ | 0.66 | 1.7-3.3[c] |
| $CaCO_3$ burial | GtC yr$^{-1}$ | 0.23 | 0.1-0.14[e] |
| opal burial | Tmol Si yr$^{-1}$ | 6.64 | 2.7-9.9[b] |
| POC burial | GtC yr$^{-1}$ | 0.24 | 0.12-0.26[c] |
| $CaCO_3$ stock | GtC | 948 | |
| opal stock | Tmol Si | 20,484 | |
| POC stock | GtC | 518 | |

pulses are prescribed in the North Atlantic as detailed in Menviel et al. (2012), also accounting for the LGM to PI change in salinity of about 1 PSU. Coral reef regrowth is implemented from 14 kyrBP onward (Vecsei and Berger, 2004) by uniformly removing the respective amount of alkalinity, carbon, and $^{13}$C from the uppermost ocean grid cells. These forcings (Fig. 3A-D) are termed standard (std.) forcings.

**2.2.2 Factorial sensitivities**

Standard transient deglacial forcings in the Bern3D model are complemented by changes in seven additional mechanisms, grouped into generic classes and introduced below (see top of Fig. 1). A set of 50 factorial experiments (Table 2), covering the same time-interval as the model run with standard transient forcings, is conducted to quantify the sensitivities of the carbon cycle to changes in these seven mechanisms. The parameters are set to their glacial values instantaneously at the beginning of
10 the glacial simulation and scaled back towards their modern values over the deglacial as illustrated in Fig. 3D,E, and F. The sensitivities of carbon cycle properties are obtained by subtracting the standard forcing run from the runs with complementary changes in the seven mechanisms. This also yields characteristic response relationships between different carbon cycle properties, e.g., changes in atmospheric $CO_2$ versus changes in whole ocean $\delta^{13}C$, for each process. The objective is to schematically represent the space of plausible property–property relationships for key proxies. In other words, the selected mechanisms are
15 assumed to be representative on the global scale for the myriad of processes influencing the carbon cycle and will be motivated below.





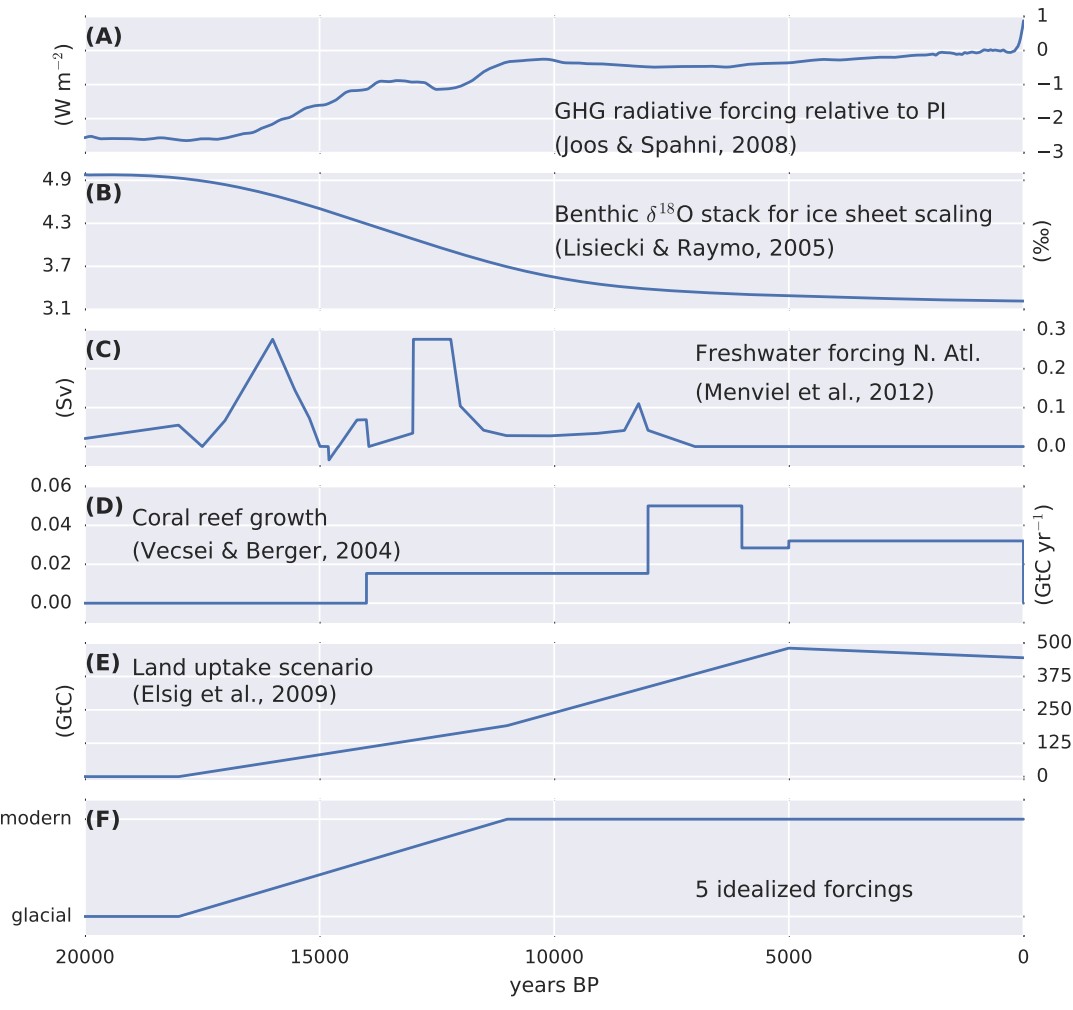

**Figure 3.** Forcings applied for the LGM to PI simulations. (A) Radiative forcing imposed by $CO_2$, $CH_4$, and $N_2O$, (B) benthic $\delta^{18}O$, used to scale ice sheet extent, (C) freshwater forcing into the North Atlantic, and (D) coral reef regrowth as applied as standard transient forcing. (E) Prescribed evolution of land biosphere carbon uptake. Cumulative land biosphere uptake is varied over the termination across factorial simulations, but kept invariant (254 GtC) over the Holocene (11–0 kyrBP) following Elsig et al. (2009); here a scenario with a total uptake of 445 GtC is shown for illustration. And (F) idealized evolution of scaling factors as prescribed in factorial experiments to vary the remineralization profile, rain ratio, Southern Ocean wind stress and gas transfer rate, or organic weathering flux on land between LGM and PI values.



**Physical Mechanisms:** Changes in ocean circulation, for example in response to altered wind stress, stratification, eddy mixing, or buoyancy forcing, affect the carbon inventory of the ocean (e.g. Adkins, 2013) by altering the cycling of organic and inorganic carbon. The Southern Ocean has been identified as an important region modulating $CO_2$ changes and a reduced Antarctic overturning circulation is invoked to, at least partly, explain low glacial atmospheric $CO_2$ (Sigman and Boyle, 2000;

Sigman et al., 2010; Fischer et al., 2010; Skinner et al., 2010; Tschumi et al., 2011; Ferrari et al., 2014; Watson et al., 2015; Roberts et al., 2016; Gottschalk et al., 2016; Jaccard et al., 2016). Under standard forcing, Southern Ocean circulation changes are weak in the Bern3D model. In this study, the circulation is modified by uniformly scaling the wind stress field over the Southern Ocean (>48°S, south of South America) (see also Tschumi et al., 2008) to invoke a reduced deep ocean ventilation. The scaling factor is varied linearly over the termination between LGM values (given in Table 2) and the PI value of unity (Fig.

3F).

Proxy reconstructions suggest a larger sea-ice extent in the Southern Ocean during glacial times (Gersonde et al., 2005; Waelbroeck et al., 2009). Sea-ice hinders air–sea gas exchange and thus exerts a large impact on atmospheric $\delta^{13}CO_2$ (Lynch-Stieglitz et al., 1995). Additionally, changes in Southern Ocean winds (Kohfeld et al., 2013) may have affected glacial Southern Ocean air–sea gas exchange rates. Here, the standard air–sea gas transfer rate is scaled (Fig. 3F; Table 2) in the Southern Ocean

to estimate related carbon cycle sensitivities. We note that changes in sea-ice and a corresponding change in air–sea gas transfer rates are reasonably well simulated by the Bern3D model. The effect of changes in ocean temperature (Waelbroeck et al., 2009) and salinity on the solubility of $CO_2$ is discussed elsewhere (Sigman and Boyle, 2000; Brovkin et al., 2007; Sigman et al., 2010; Menviel et al., 2012) and is part of the std. forcing.

**Oceanic carbonate:** The partial pressure of $CO_2$ in the ocean depends on alkalinity and processes that alter surface ocean alkalinity thus influence atmospheric $CO_2$. The sedimentary burial of $CaCO_3$ removes twice as much alkalinity as carbon from the ocean leading to oceanic $CO_2$ outgassing. Increased $CaCO_3$ burial over the deglacial has been put forward as the coral reef hypothesis (Berger, 1982) to explain increases in atm $CO_2$. Vecsei and Berger (2004) reconstructed coral reef growth history and provided a lower limit estimate of about 380 GtC for the amount of shallow water carbonate deposition over the deglacial

period. Other estimates of $CaCO_3$ deposition amount to 1,200 GtC or even more (Milliman, 1993; Kleypas, 1997; Ridgwell et al., 2003). This yields a large range of possible scenarios with substantial impacts on atmospheric $CO_2$. Here, we scale the reconstructed deposition history based on Vecsei and Berger (2004) (Fig. 3D) by applying a constant scaling.

Potential changes in the rain ratio ($CaCO_3$:$C_{org}$) affect both alkalinity and $CO_2$ (e.g. Berger and Keir, 2013; Archer and Maier-Reimer, 1994; Sigman et al., 1998; Archer et al., 2000; Matsumoto et al., 2002; Tschumi et al., 2011). An increase

in the export of $CaCO_3$ relative to $C_{org}$ from surface waters has similar consequences for $CO_2$ as shallow water carbonate deposition (Sigman and Boyle, 2000). Here, we vary the global rain ratio of 0.083 (see also Appendix A; Fig. 3F; Table 2).

**Oceanic organic matter:** There is a broad range of mechanisms, in addition to circulation changes, that affect the cycling of organic matter within the ocean and the whole ocean nutrient inventories, and thereby surface ocean nutrient and DIC

concentrations as well as atmospheric $CO_2$. Several studies have suggested changes in the remineralization length scale of



**Table 2.** Overview of mechanisms and parameters considered in factorial sensitivity experiments. Bold entries mark the standard parameter values. Values represent scaling factors in case of a standard parameter equal to unity.

| Mechanisms | Parameter values |
|---|---|
| Southern Ocean wind stress | 0.4, 0.5, 0.6, 0.7, 0.8, 0.9, **1.0** |
| Southern Ocean gas exchange flux | 0.4, 0.5, 0.6, 0.7, 0.8, 0.9, **1.0**, 1.1, 1.2 |
| Export rain ratio | 0.045, 0.05, 0.055, 0.063, 0.068, 0.073, **0.083**, 0.088, 0.093, 0.098 |
| Shallow water carbonate deposition | 0, 0.5, **1**, 1.5, 2, 3, 4 |
| % change towards linear remineralization profile | **0**, 10, 20, 40, 60, 80, 100 |
| Organic weathering flux | 0.5, 0.6, 0.7, 0.8, 0.9, **1.0**, 1.2, 1.4, 1.6, 2.0 |
| Land biosphere uptake (GtC) | -500, -200, **0**, 445, 890, 1335 |

organic carbon due to colder ocean temperatures (see e.g. Matsumoto, 2007; Menviel et al., 2012; Roth et al., 2014). A deeper remineralization of organic matter leads to the removal of nutrient and DIC from the shallow subsurface ocean and, via ocean–sediment interactions, to globally reduced nutrient and increased alkalinity inventories (Menviel et al., 2012; Roth et al., 2014), resulting in a decrease in atmospheric $CO_2$ (Matsumoto, 2007; Menviel et al., 2012) amplified through ocean–

sediment interactions (Roth et al., 2014). Weathering-burial feedbacks also lead to an amplification of changes in atmospheric and oceanic $\delta^{13}C$ as shown by Tschumi et al. (2011) and Roth et al. (2014). Here, the remineralization profile in the upper 2 km is varied between the standard Martin curve (Eq. A1) and a linear profile.

    Oceanic organic carbon and nutrient inventories are further affected by changes in the weathering flux compensating for burial of organic matter (Cartapanis et al., 2016, 2018). A negative balance between the riverine input of $PO_4$, DIC, $DI^{13}C$, and

Alk, originally buried as organic material, and sedimentary burial of particulate organic matter leads to a transient decrease in atmospheric $CO_2$ (Cartapanis et al., 2016) and also affects atmospheric and marine $\delta^{13}C$. Here, we varied the input of $PO_4$, DIC, $DI^{13}C$, and Alk that compensates for the burial of particulate organic material in the sediment in factorial experiments (Fig. 3F; Table 2). This variation in the input will be referred to as changes in the organic weathering flux.

    Another widely discussed mechanism relates to iron fertilization in the Southern Ocean (Martin, 1990; Archer et al., 2000;

Sigman and Boyle, 2000; Parekh et al., 2008; Sigman et al., 2010; Fischer et al., 2010; Martinez-Garcia et al., 2014). The effect of iron fertilization on oceanic carbon and $\delta^{13}C_{DIC}$ is similar to the mechanisms outlined above and is thus not explicitly considered further. Other processes such as changes in the ratio of nutrient to carbon uptake by marine organisms (Heinze et al., 2016) or changes in the association of POC with ballast minerals Armstrong et al. (2002) may have affected the efficiency of the marine biological cycle in reducing carbon in surface water and the atmosphere.

    **Land biosphere carbon inventory:** Changes in land biosphere carbon stock are prescribed following the evolution illustrated in Fig. 3E. We assume that the change in land carbon stock over the Holocene is well constrained and amounts to an



**Table 3.** Overview of the four proxy-based constraints representing change ($\Delta$) between PI minus LGM. LGM refers to the period 20 to 19 kyrBP for the ice core and to the period 23 to 19 kyrBP for the ocean sediment data. Similarly, PI refers to 500 to 200 yrBP for the ice core data and to 6,000 to 200 yrBP and to 8 to 6 kyrBP for marine $\delta^{13}C$ and $CO_3^{2-}$, respectively. The assumed uncertainty range in $\Delta CO_2$ is considered to be wider than the uncertainty from the ice core data. This is to avoid overfitting and taking into account uncertainties in the emulator introduced in Section 2.2.4.

| Constraint | uncertainty range | references |
|---|---|---|
| $\Delta CO_2$ | 80 to 100 ppm | see text |
| $\Delta\delta^{13}C_{atm}$ | 0.1±0.05 ‰ | Schmitt et al. (2012) |
| $\Delta\delta^{13}C_{DIC}$ | 0.34±0.19 ‰ | Peterson et al. (2014) |
| $\Delta CO_3^{2-}$ | -6 to 5 µmol kg$^{-1}$ | see text |

uptake of 290 GtC between 11–5 kyrBP and a release of 36 GtC thereafter (Elsig et al., 2009). Scenarios with different cumulative land carbon change imply a corresponding uptake or release of terrestrial carbon during the last glacial termination (18–11 kyrBP). The $\delta^{13}C$ signature of terrestrial carbon is set to -24 ‰. Additional sensitivity tests varying the $\delta^{13}C$ signature between -23 ‰ and -25 ‰ are conducted and will be discussed in Section 4.

### 2.2.3 Data constraints for deglacial carbon and carbon-isotope changes

We consider four globally relevant observational constraints (Table 3) to estimate the PI-LGM change in the land biosphere carbon inventory, $\Delta$land. These are the PI-LGM difference ($\Delta$) in (i) atmospheric $CO_2$ ($\Delta CO_2$), (ii) atmospheric $\delta^{13}C(CO_2)$ ($\Delta\delta^{13}C_{atm}$), (iii) mean ocean $\delta^{13}C(DIC)$ ($\Delta\delta^{13}C_{DIC}$), and (iv) deep (equatorial) Pacific carbonate ion concentration ($\Delta CO_3^{2-}$) (Table 3). The carbon isotope constraints are directly relevant for the isotopic mass balance used to infer $\Delta$land. The carbonate

ion concentration difference constrains changes in the ocean's alkalinity budget, and the change in atmospheric $CO_2$ constrains the integrated effect of deglacial carbon cycle changes. We allow for a relatively wide uncertainty range of 20 ppm for $\Delta CO_2$, which is admittedly wider than the proxy uncertainty, to avoid an overfitting of model outputs. The $\Delta\delta^{13}C_{atm}$ range for the constraint is 0.1 ±0.05 ‰ and based on ice core measurements (Schmitt et al., 2012). Peterson et al. (2014) compiled 480 benthic foraminiferal records and report a whole-ocean PI minus LGM change in $\delta^{13}C_{DIC}$ of 0.34±0.19 ‰ and we use their

range (0.15 to 0.53 ‰) as constraint. Proxy reconstructions of $\Delta CO_3^{2-}$ based on measured B/Ca ratios in foraminifera show little change from LGM to PI in the equatorial deep Pacific (Yu et al., 2013). Based on a different approach, Qin et al. (2017) also conclude on negligible change in $\Delta CO_3^{2-}$ in the western tropical Pacific across the last glacial termination. In a recent study, Luo et al. (2018) report, based on CaCO_3 reconstructions in the South China Sea, that carbonate ion concentrations in the mid-depth Pacific hardly changed from LGM to PI. All these studies point to a small change in $\Delta CO_3^{2-}$, providing a further

constraint on deglacial simulations. However, the uncertainty in these reconstructions is of the same order as the change itself. The target range for $\Delta CO_3^{2-}$ is chosen here from -6 to 5 µmol kg$^{-1}$.





#### 2.2.4 Emulator and Monte Carlo setup

Sensitivities of the four target proxies ($\Delta CO_2$, $\Delta\delta^{13}C_{atm}$, $\Delta\delta^{13}C_{DIC}$, and $\Delta CO_3^{2-}$) to changes in the selected seven forcings are computed based on 50 factorial experiments. For each forcing, $i$, and related parameter change $\Delta p_i$ (Table 2) the corresponding sensitivity, i.e., the change in target per unit change in parameter, is computed. These discrete sensitivities are fitted linearly or quadratically for each target, $T$, and forcing to get a continuous function for the sensitivity, $S_i^T(\Delta p_i)$. These sensitivity functions are used to build a simple representation to compute changes in target variables for a combination of changes in parameters:

$$\Delta T(\Delta p_1, ..., \Delta p_7) = \sum_{i=1}^{7} \Delta p_i \times S_i^T(\Delta p_i) \tag{1}$$

In order to evaluate the assumed linear additivity of the seven mechanisms, a 60 member ensemble of parameter combinations is calculated both with the Bern3D model and Eq. 1 (Fig. 1). The seven parameters are sampled uniformly using Latin Hypercube Sampling (LHS; McKay et al., 1979). The results from the Bern3D model and Eq. 1 are regressed against each other for each target variable (Fig. C1). The corresponding linear fit is used to account for non-additive behavior when combining forcings. This then yields the following emulator:

$$\Delta T(\Delta p_1, ..., \Delta p_7) = a + b \times \left( \sum_{i=1}^{7} \Delta p_i \times S_i^T(\Delta p_i) \right), \tag{2}$$

where $a$ is the offset and $b$ the slope of the linear fit as given in Fig. C1. The root mean square deviation between the emulator (Eq. 2) and the Bern3D model is 8.8 ppm for $\Delta CO_2$, 0.06 ‰ for $\Delta\delta^{13}C_{atm}$, 0.02 ‰ for $\Delta\delta^{13}C_{DIC}$, and 4 µmol kg$^{-1}$ for $\Delta CO_3^{2-}$. The seven parameters of the LGM to PI sensitivity experiments are sampled uniformly, by drawing random samples over their employed ranges (Table 2) to generate a 500,000 member Monte Carlo ensemble. The range for $\Delta$land was extended from 1,335 GtC to 1,500 GtC in order to cover the range of estimates. This is justified by the very linear relationship of the four targets to changes in $\Delta$land (e.g. Fig. 7). The above emulator is used for each of these members to estimate the corresponding changes in the four targets. Members which yield changes in the targets that lie outside the defined ranges (Table 3) are excluded from the analysis; all remaining members are used to calculate median and confidence ranges in $\Delta$land and other variables of interest.

## 3 Results

### 3.1 Pulse experiments

We start by investigating how atmospheric $CO_2$ and $\delta^{13}C$ signatures evolve in response to a negative pulse emission of light carbon ($\delta^{13}C$=-24 ‰). First, we apply a closed system, which is including only the atmosphere, ocean, and land biosphere components. The removal of 100 Gt of light carbon from the atmosphere results in an initial decline in atmospheric $CO_2$





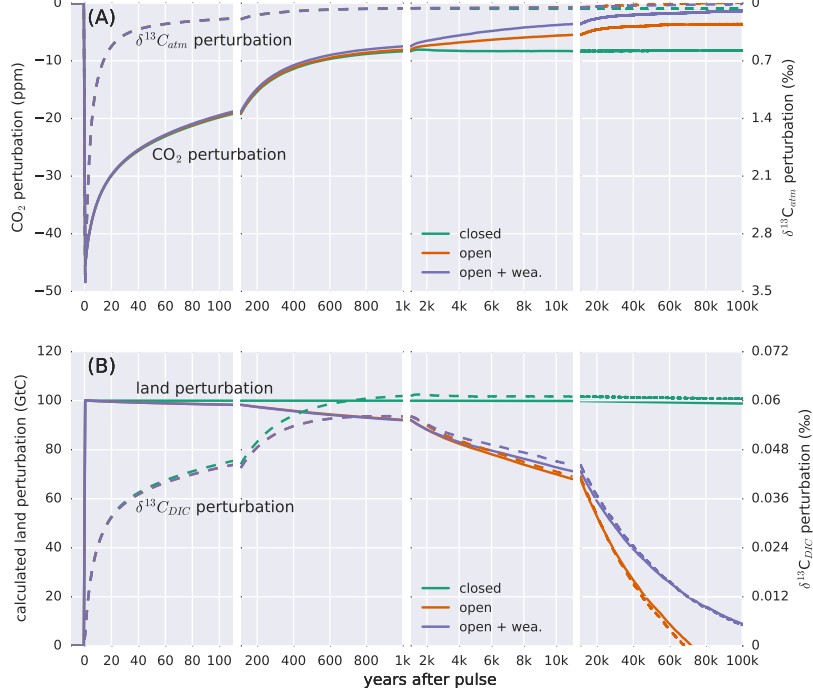

**Figure 4.** Temporal evolution of the perturbation in (A) atmospheric $CO_2$ and $\delta^{13}C_{atm}$ (inverted y-axis), and (B) the calculated perturbation in terrestrial carbon storage following Eq. A2 and A3 and $\delta^{13}C_{DIC}$ anomalies for a pulse uptake of 100 GtC by the land in a closed (green), open (orange), and open system with enabled weathering feedback (purple). Solid lines refer to left and dashed lines to right y-axis in both panels.

followed by a typical impulse–response recovery. Atmospheric $CO_2$ recovers as the ocean starts to outgas $CO_2$ (Fig. 4). In a closed system, atmospheric $CO_2$ reaches a new equilibrium after about 2 kyr, at 8 ppm below the initial steady state. As light carbon is removed, the remaining carbon in the ocean and atmosphere becomes relatively enriched in $^{13}$C. Initially, the $\delta^{13}C_{atm}$ perturbation is removed much faster than the atmospheric $CO_2$ perturbation as exchange with the ocean and land

5     biosphere dilutes the imposed isotopic perturbation (Fig. 4A). The oceanic $\delta^{13}$C increases and equilibrates after about 2 kyr. The resulting change for the new equilibrium in $\delta^{13}$C amounts to 0.066 ‰ for the atmosphere, 0.060 ‰ for the ocean, and 0.065 ‰ for the 4-box land biosphere. The mass balance between ocean, land and atmosphere is maintained such that the inferred land perturbation (calculated using Eq. A2 and A3) corresponds to the prescribed removal of 100 GtC (Fig. 4B).

10     In the second, open system experiment, sediments are included. The evolution of atmospheric $CO_2$ is similar to the closed system except that $CaCO_3$ compensation leads to a further recovery of atmospheric $CO_2$ until equilibrium is reached after ~60 kyr. $CO_2$ equilibrates at around 3.5 ppm below the initial steady state. For the first kyr, the evolution of $\delta^{13}$C in the atmosphere and ocean is comparable to the closed system experiment but differs thereafter. In both the atmosphere and ocean, the $\delta^{13}$C perturbation is decreasing compared to the open system experiment on a multi-millenial timescale (see dashed lines in Fig.



4B), related to imbalances in the sedimentation (deposition minus redissolution/oxidation at the ocean–sediment interface) and weathering fluxes of POC and $CaCO_3$.

The $^{13}C$ budget in the model may change as a result of imbalances between input and burial fluxes of POC and $CaCO_3$, or from changes in the $\delta^{13}C$ signatures of the respective fluxes. Carbon–climate feedbacks over the course of the experiment

cause changes in temperature and ocean circulation, which affect the export of POC and $CaCO_3$. The removal of light terrestrial carbon increases the $\delta^{13}C$ signature of surface waters and by that the $\delta^{13}C$ signatures of exported POC and $CaCO_3$. Burial fluxes of POC further depend on $O_2$ concentrations and burial fluxes of $CaCO_3$ on $CO_3^{2-}$ concentrations which evolve over the course of the simulation. As a result, the cumulative sedimentation-weathering imbalance for the open system setting for the first 10 kyr after the pulse uptake amounts to a removal of 48.5 GtC from the system. The mean $\delta^{13}C$ signature of the burial

flux, combining mean $CaCO_3$ ($\delta^{13}C_{CaCO_3}$=2.9 ‰) and $C_{org}$ ($\delta^{13}C_{POC}$=-20.2 ‰) signatures, over these first 10 kyr amounts to -8.7 ‰ and is thus larger than the mean $\delta^{13}C$ signature of the weathering input ($\delta^{13}C$=-9.1 ‰, determined after the spin-up). Hence, $^{13}C$ is lost from the atmosphere–ocean system and $\delta^{13}C_{DIC}$ is decreasing. The resulting difference in the $\delta^{13}C_{DIC}$ perturbation between open and closed system causes an erroneous mass balance inference for the terrestrial biosphere, when applying conventional closed system equations (Eq. A2 and A3). On a timescale of 10 kyr, underestimation of the terrestrial

carbon inventory change amounts to 30 %. The error increases as times evolves. The contributions to this decrease in $\delta^{13}C$ are 65 % and 35 % for changes in the cycling of POC and $CaCO_3$, respectively (as calculated 10 kyr after the pulse).

In the third pulse experiment, changes in global mean air temperature and atmospheric $CO_2$ concentration affect weathering fluxes of $CaCO_3$ and $CaSiO_3$ following Colbourn et al. (2013). The implemented feedbacks cause a reduction in modeled

weathering fluxes in response to decreased atmospheric $CO_2$ and temperature over the course of the experiment. Decreased weathering fluxes of $CaCO_3$ and $CaSiO_3$ lead to a decline in alkalinity supplied to the ocean resulting in a net transfer of carbon to the atmosphere. This further reduces the initial $CO_2$ perturbation (Fig. 4A). The mean $\delta^{13}C$ value related to weathering is -9.1 ‰ thereby leading to a slightly higher $\delta^{13}C_{DIC}$ perturbation at 10 kyr compared to the open system experiment with fixed weathering fluxes. The error related to the calculated land perturbation is comparable on a 10 kyr time horizon, and increases

less as time progresses further compared to the open system setup with fixed weathering.

From these pulse experiments it becomes clear that sediment and weathering interactions affect the carbon and carbon isotope budgets on deglacial timescales. The contribution to differences in the carbon isotopic budget between the closed and open system is dominated by changes in POC cycling and its associated $\delta^{13}C$ signature, while changes in the $CaCO_3$ cycle

play a smaller, but non-negligible, role on the isotopic budget. Overall we find that calculations of $\Delta$land that rely on the closed system isotopic mass balance approach are underestimated.

## 3.2 Ensemble approach to constrain $\Delta$land

We now turn to estimate $\Delta$land. We recognize the deglacial reorganization of the ocean, as represented in the standard model setup and by the seven deglacial carbon cycle mechanisms and the four observational targets within an open system model





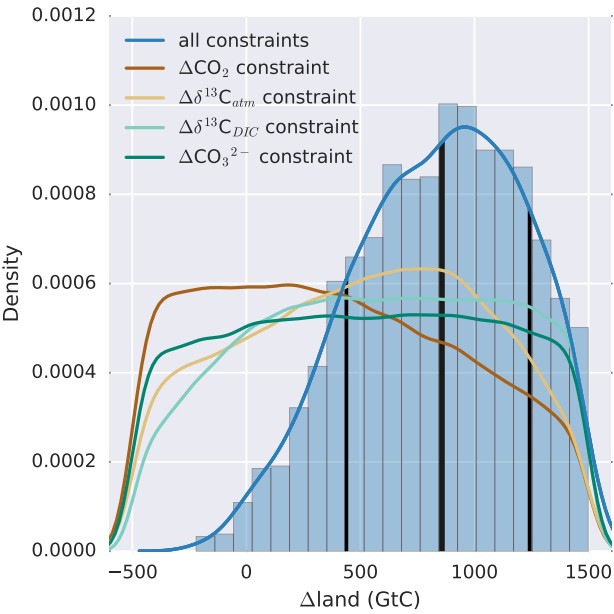

**Figure 5.** Histogram (blue bars, normalized) and kernel density estimate (blue line) of $\Delta$land where parameter combinations fulfill all four data-based constraints. Vertical thick black line indicates the median $\Delta$land of $\sim$850 GtC and thin black lines show the $\pm 1\sigma$ range. Remaining four lines show kernel density estimates of $\Delta$land when considering one data-based constraint at a time.

framework. Applying the Bern3D emulator (Eq. 2) to our 500,000 member parameter ensemble, we find that the prior range in $\Delta$land, ranging from -500 to 1,500 GtC, is constrained to $\sim$450 to $\sim$1,250 GtC ($\pm 1\sigma$ range, see Fig. 5) in the light of the four observational targets. The constrained median amounts to $\sim$850 GtC, and almost no ensemble members which fulfill all four observational constraints are found for negative $\Delta$land values. The constrained ensemble thus clearly suggests a positive

$\Delta$land. When considering single constraints only, $\Delta$land values cover the whole range of sampled values almost uniformly (Fig. 5). Considering either $\Delta\delta^{13}C_{DIC}$ or $\Delta\delta^{13}C_{atm}$ together with $\Delta CO_2$ as constraints moves the $\Delta$land distribution towards the constrained distribution (blue line in Fig. 5; not shown). The incorporation of multiple constraints is thus essential in narrowing down the $\Delta$land estimate.

With the chosen ranges in the seven parameters, our unconstrained ensemble yields a very large spread in the four target variables. $\Delta\delta^{13}C_{DIC}$ ranges from -0.77 to 1.25 ‰; $\Delta\delta^{13}C_{atm}$ from -0.9 to 1.32 ‰; $\Delta CO_2$ from -107 to 181 ppm; and $\Delta CO_3^{2-}$ from -49 to 42 µmol kg$^{-1}$. Within the optimization procedure, we specifiy that only defined, observation-based ranges of these target variables are allowed (Section 2.2.3), and that all four of these constraints have to be met at the same time. This yields specific distributions of the target variables within our constrained ensemble (Fig. 6, magenta lines). The distribution of

$\Delta\delta^{13}C_{atm}$ values, for instance, is almost uniform within the defined observational range (Fig. 6B), while $\Delta CO_2$, $\Delta CO_3^{2-}$, and




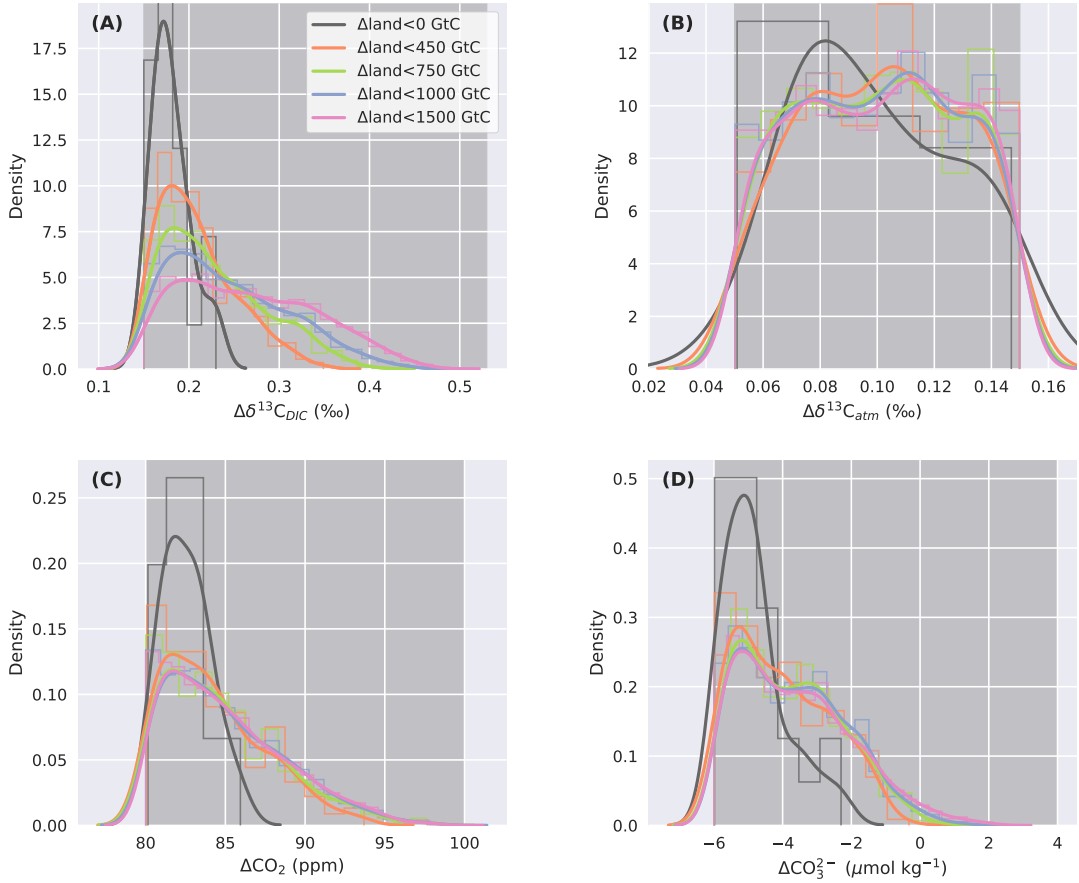

**Figure 6.** Histogram (thin lines, normalized) and kernel density estimate (thick lines) of (A) $\Delta\delta^{13}C_{DIC}$, (B) $\Delta\delta^{13}C_{atm}$, (C) $\Delta CO_2$, and (D) $\Delta CO_3^{2-}$ values for parameter combinations calculated with the emulator that fulfill the observational constraints. Different colors in all four panels correspond to sub-samples divided along $\Delta$land values that are smaller than 1,500, 1,000, 750, 450, and 0 GtC. Darker grey shading indicates the range of the respective observational constraint.

$\Delta\delta^{13}C_{DIC}$ values cluster towards the lower end of the specified target ranges (Fig. 6C,D,A). This is a result of the underlying carbon cycle–climate processes as tested within Bern3D. For example, it remains difficult to model a PI-LGM $\Delta CO_2$ of ∼90 ppm under the constraint of meeting the other three observational targets. Most ensemble members represent a $\Delta CO_2$ of ∼82 ppm (Fig. 6C). Higher $\Delta CO_2$ are present in the unconstrained ensemble, but are generally associated with even more negative 5 $\Delta CO_3^{2-}$ (not shown), which is outside the defined target range for $\Delta CO_3^{2-}$.

As a further test of our optimization scheme and the resulting representation of our target variables within the constrained ensemble, we consider different subsets of our ensemble where the range of the input parameter $\Delta$land is modified. We define four such sets where $\Delta$land is allowed to be maximally 1,000, 750, 450, and smaller than 0 GtC (Fig. 6, blue, green, orange, and grey lines). Only one target variable, $\Delta\delta^{13}C_{DIC}$, is sensitive to this restriction within the constrained ensemble and positive



$\Delta$land. The distributions of $\Delta\delta^{13}C_{atm}$, $\Delta CO_2$, and $\Delta CO_3^{2-}$ are almost identical for all subsets of $\Delta$land with the exception of $\Delta$land smaller than 0 GtC. Hence, these three targets do not per se discriminate between smaller and larger estimates of positive $\Delta$land. In contrast, restricting $\Delta$land to smaller values shifts the realized $\Delta\delta^{13}C_{DIC}$ towards lower values and the mean of the constrained distribution shifts away from the mean observational estimate of 0.34 ‰ (Fig. 6A). For example,

$\Delta\delta^{13}C_{DIC}$ is always below 0.35 ‰ for $\Delta$land below 450 GtC and always below 0.23 ‰ for negative $\Delta$land estimates. Thus, not only very few process combinations yield a negative $\Delta$land in our 500,000 member ensemble (Fig. 5), these solutions are in addition biased low in $\Delta\delta^{13}C_{DIC}$. Also, it is visible that for negative $\Delta$land not only $\Delta\delta^{13}C_{DIC}$ values are at the very low end of the observational range, but the same holds true for $\Delta CO_2$ and $\Delta CO_3^{2-}$ (Fig. 6B,C). Visual inspection of Fig. 6A shows a number of samples cluster around the best guess estimate of $\Delta\delta^{13}C_{DIC}$, except when $\Delta$land is restricted to below 450 GtC.

Taken together, our framework strongly suggests that the land biosphere sequestered carbon over the deglaciation and that the sequestered amount is likely larger than 450 GtC.

### 3.3 Carbon and $\delta^{13}C$ changes in LGM to PI sensitivity simulations

We now discuss the results obtained with the Bern3D model for the transient LGM to PI simulations. Forcing the model into an LGM state by adjusting GHG radiative forcing, ice sheet extent and albedo, orbital forcing, and coral reef regrowth (standard

simulation) leads to a shoaling and slight slowdown of the Atlantic meridional overturning circulation of about 4.4 Sv and a global cooling of the ocean and the atmosphere of about 1.4 °C and 3 °C, respectively. The oceanic carbon inventory increases at the expense of the atmosphere, yielding a higher $CO_3^{2-}$ concentration during the glacial and a PI-LGM $\Delta CO_2$ of 27.8 ppm (Fig. 7). $\Delta\delta^{13}C_{atm}$ and $\Delta\delta^{13}C_{DIC}$ show little change between LGM and PI. By applying the standard transient forcings alone, neither the $\Delta\delta^{13}C$ constraints in the atmosphere and ocean nor the $\Delta CO_2$ constraint is met, calling for the consideration of

additional processes.

#### 3.3.1 Sensitivity of carbon cycle processes to changes in additional deglacial processes

The mean $\delta^{13}C$ signature of the ocean–atmosphere system changes in response to potential deglacial processes (Fig. 7A-B). For simplicity, we focus on the change in mean ocean $\delta^{13}C_{DIC}$ (Fig. 7A) as the ocean holds about 20 times more carbon than the atmosphere and interactive land biosphere together. The factorial simulations reveal a $\Delta\delta^{13}C_{DIC}$ (PI-LGM) between

-0.3 to more than 0.6 ‰ when halving or doubling the weathering flux of organic material, respectively. Prescribed changes in Southern Ocean (SO) wind stress, inducing changes in ocean circulation, the organic matter remineralization profile, and in the rain ratio lead to a change in the ocean signature of about -0.2 ‰, while changes in the SO air–sea gas transfer rate and in coral reef growth have a small influence on $\Delta\delta^{13}C_{DIC}$. In summary, not only changes in terrestrial carbon storage but several processes, that were potentially important for the reorganisation of the deglacial carbon cycle, influence the mean ocean $\delta^{13}C$

signature in comparison to the observed deglacial change of 0.34±0.19 ‰. Their influence cannot be ignored when estimating $\Delta$land based on $\delta^{13}C$.

Reconstructions of the change in $\delta^{13}C$ in the atmosphere and ocean differ in magnitude. The change in the atmosphere (0.1±0.05 ‰) is about one third of the oceanic change (0.34±0.19 ‰). Changes in the organic weathering flux and terrestrial





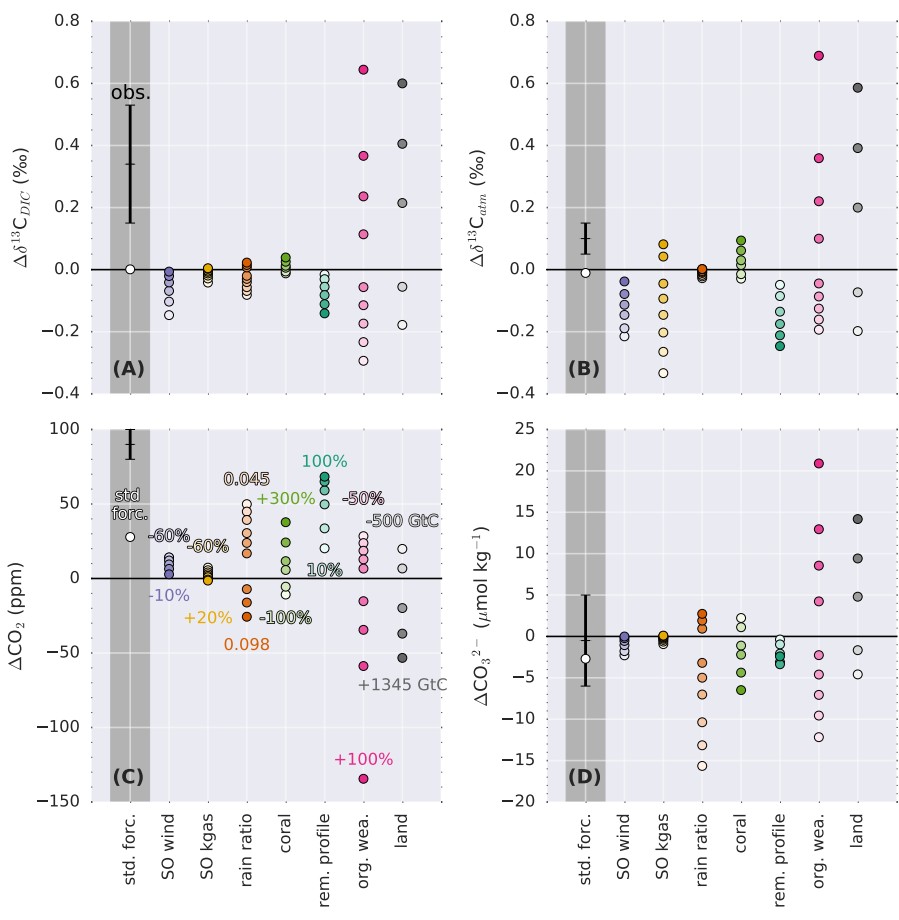

**Figure 7.** Sensitivity of (A) $\Delta\delta^{13}C_{DIC}$, (B) $\Delta\delta^{13}C_{atm}$, (C) $\Delta CO_2$, and (D) $\Delta CO_3^{2-}$ to changes in Southern Ocean wind stress (SO wind), Southern Ocean gas exchange (SO kgas), export rain ratio, shallow water carbonate deposition (coral), remineralization profile (rem. profile), organic weathering flux (org. wea.), and land carbon uptake (land). Sensitivities are shown relative to the std. forcings (std. forc., white dots). Black crosses with errorbars show estimates based on measurements ($\Delta CO_2$: Petit et al. (1999); Monnin et al. (2001); Siegenthaler et al. (2005); Lüthi et al. (2008); Köhler et al. (2017); $\Delta\delta^{13}C_{atm}$: Schmitt et al. (2012); $\Delta CO_3^{2-}$: Yu et al. (2013); Qin et al. (2017); Luo et al. (2018); $\Delta\delta^{13}C_{DIC}$: Peterson et al. (2014)). Dots in dark grey shading show absolute and in light grey shading relative (to std. forcings) values. $\Delta$ indicates the PI-LGM difference.





carbon storage affect $\Delta\delta^{13}C_{DIC}$ and $\Delta\delta^{13}C_{atm}$ in an identical way. Changes in SO wind stress, rain ratio, coral reef growth, and the remineralization of organic material have a similar, though not completely identical, influence on the two isotope variables. In contrast, changes in the SO air–sea gas transfer rate alter $\Delta\delta^{13}C_{atm}$, while hardly changing $\Delta\delta^{13}C_{DIC}$. The strong sensitivity of $\delta^{13}C_{atm}$ to changes in SO gas transfer rate is due to the large disequilibrium between the atmosphere

and surface ocean. This makes this process, in addition to temperature driven changes in fractionation, potentially important in setting atmospheric $\delta^{13}C$ independently from mean ocean $\delta^{13}C_{DIC}$ change (see also Fig. C2B).

Considering $\Delta CO_2$ and $\Delta CO_3^{2-}$, the changes in investigated mechanisms significantly influence atmospheric $CO_2$ and deep Pacific carbonate ion concentration (with the exception of air–sea gas transfer rate). A general relationship becomes apparent. A positive change in $\Delta CO_2$ is always accompanied by a negative change in $\Delta CO_3^{2-}$ and vice versa (Fig. 7C-D). Thus, the

explanation of the observed deglacial increase in $CO_2$ of $\sim$90 ppm requires a deglacial decrease in deep Pacific carbonate ion concentration (Fig. 6D), at least in our model setting and considering the mechanisms implemented in the standard and the factorial simulations.

### 3.3.2 Response relationships between different carbon cycle properties

The results for the target variables are plotted against each other to gain further insight on the relative importance of the different

mechanisms to meet the observational constraints (Fig. 8). This yields for each target pair and mechanism a characteristic slope. For example, this slope is negative and equates to about -90 ppm per ‰ for the pair $\Delta CO_2$ and $\Delta\delta^{13}C_{DIC}$ and the mechanism land carbon storage (grey line in Fig. 8A). Roughly similar slopes are found for variations of the organic weathering flux, SO wind stress, and SO air–sea gas transfer rate. Though, the overall impact of SO air–sea gas transfer rate changes on these targets is small. Visual inspection of Fig. 8A reveals that variations in these four processes alone cannot prompt the model results to

agree with these two observational constraints. The slopes are too small in magnitude and either $\Delta CO_2$ or $\Delta\delta^{13}C_{DIC}$ remains outside the observational range also when varying these four processes in combination. The four processes are effective in varying $\Delta\delta^{13}C_{DIC}$ but relatively ineffective in varying $\Delta CO_2$. Processes that have a much larger impact on $\Delta CO_2$ than on $\Delta\delta^{13}C_{DIC}$ are related to the $CaCO_3$ cycle (coral reef growth and rain ratio) and changes in the upper ocean remineralization of particulate organic matter. This implies that variations in $\Delta$land, organic weathering, and ocean circulation (SO wind) are

required to meet the $\Delta\delta^{13}C_{DIC}$ target, while variations in coral reef regrowth, rain ratio, and upper ocean remineralization depth are necessary to meet the $\Delta CO_2$ target. Indeed, the median changes for the parameters in the observationally constrained ensemble (Fig. C2) imply a deeper remineralization of organic matter and a reduced rain ratio at LGM than PI and a three to four times higher amount of $CaCO_3$ deposition during coral reef regrowth than applied in the standard run. These, together with a reduced SO wind stress at LGM, all contribute to a deglacial increase in $CO_2$. The associated decrease in $\Delta\delta^{13}C_{DIC}$ from

these changes is in most Monte Carlo realizations more than compensated by a deglacial increase in land carbon inventory, and in many realizations by an increase in the organic weathering flux (Fig. C2, 8F-G). In general, the smaller the change in $\Delta$land, the larger the increase required in the flux of organic weathering.

Slopes for the target pair $\Delta\delta^{13}C_{atm} - \Delta CO_2$ resemble the $\Delta\delta^{13}C_{DIC} - \Delta CO_2$ pair (Fig. 8A-B). Variations in SO air–sea gas transfer rate, SO wind stress, $\Delta$land, and organic weathering flux show relatively high sensitivity for $\Delta\delta^{13}C_{atm}$ and low



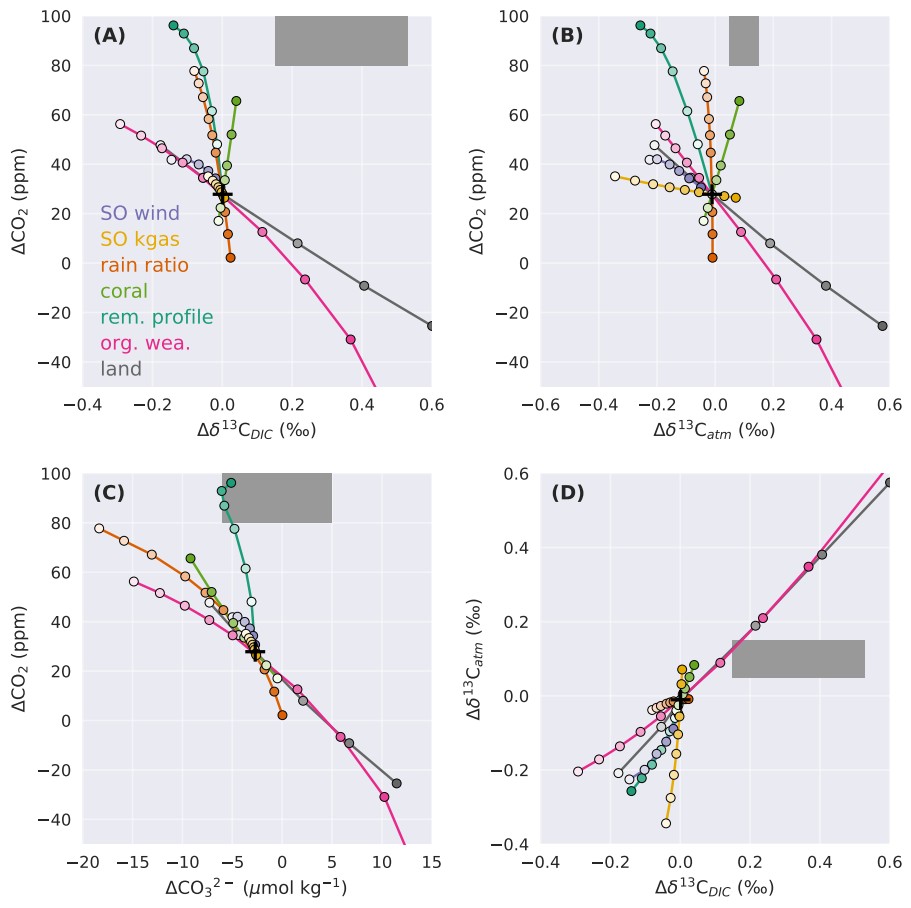

**Figure 8.** $\Delta CO_2$ for defined sensitivity experiments plotted against (A) $\Delta\delta^{13}C_{atm}$, (B) $\Delta\delta^{13}C_{DIC}$, (C) $\Delta CO_3^{2-}$, and (D) $\Delta\delta^{13}C_{atm}$ vs. $\Delta\delta^{13}C_{DIC}$. Black crosses indicate results with std. forcings alone and dark grey shadings show ranges for data based constraints ($\Delta CO_2$: Petit et al. (1999); Monnin et al. (2001); Siegenthaler et al. (2005); Lüthi et al. (2008); Köhler et al. (2017); $\Delta\delta^{13}C_{atm}$: Schmitt et al. (2012); $\Delta CO_3^{2-}$: Yu et al. (2013); Qin et al. (2017); Luo et al. (2018); $\Delta\delta^{13}C_{DIC}$: Peterson et al. (2014)). For the direction of change in the forcings see Fig. 7. Note that in (A) changes in $\Delta$land that yield a positive $\Delta CO_2$ are overlain by the organic weathering flux line.



sensitivity for $\Delta CO_2$, whereas coral reef regrowth, rain ratio, and upper ocean remineralization changes exert a large impact on $\Delta CO_2$ and affect $\Delta\delta^{13}C_{atm}$ little (Fig. 8B). In contrast to the $\Delta\delta^{13}C_{DIC} - \Delta CO_2$ target pair (Fig. 8A), the slopes for the $\Delta\delta^{13}C_{atm} - \Delta CO_2$ pair do not fall on roughly two characteristic slopes but vary across processes (Fig. 8B).

The clear relationship between oceanic alkalinity and oceanic carbon uptake (Fig. 7C-D) is also visible in Fig. 8C. Higher

alkalinity during the glacial (negative $\Delta CO_3^{2-}$) generally results in a larger drawdown of atmospheric $CO_2$ (positive $\Delta CO_2$). An interesting exception is that a modest deepening of the remineralization of organic matter in the upper ocean leads to large changes in atmospheric $CO_2$, while at the same time hardly changing deep Pacific $CO_3^{2-}$, making it a potentially important process to fulfill the $\Delta CO_2$ target without shifting $\Delta CO_3^{2-}$ away from the observationally constrained range (Fig. 8C). Accordingly, the remineralization depth of organic matter is deeper at the LGM than PI in all Monte Carlo realizations (Fig.

C2E).

Next, we consider the $\Delta\delta^{13}C_{DIC} - \Delta\delta^{13}C_{atm}$ target pair (Fig. 8D). The difficulty in simultaneously fulfilling the atmospheric and oceanic $\delta^{13}C$ constraints with a negative $\Delta$land is becoming obvious. With negative $\Delta$land, $\Delta\delta^{13}C_{atm}$ and $\Delta\delta^{13}C_{DIC}$ both are moved towards negative values relative to std. forcings. Therefore, the isotopic constraints could only be fulfilled with increased organic weathering flux such that $\Delta\delta^{13}C_{atm}$ and $\Delta\delta^{13}C_{DIC}$ both are moved towards positive values

relative to the std. forcings (Fig. 8D). Considering the $\Delta CO_2$ and $\Delta CO_3^{2-}$ constraints (Fig. 8C) implies that negative $\Delta$land can only be offset by increased organic weathering or rain ratio. However, reaching a $\Delta CO_2$ of 80–100 ppm remains difficult for negative $\Delta$land when combining information from all panels of Fig. 8. This explains why only very few solutions with negative $\Delta$land and no solutions with $\Delta$land smaller than -220 GtC are found in the Monte Carlo simulations (Section 3.2).

### 3.3.3 Sensitivity of the spatial distribution of $\Delta\delta^{13}C_{DIC}$ to changes in deglacial processes

In this section, we describe the spatial distribution of changes in $\Delta\delta^{13}C_{DIC}$. Despite little change on global average, the reference run with standard LGM forcings (Fig. 9A) shows substantial spatial changes in $\Delta\delta^{13}C_{DIC}$ from internal reorganization linked to ocean circulation changes. Reduced glacial ocean ventilation as evidenced in changes in ideal age leads to the accumulation of light carbon from marine export productivity in the ocean interior. Circulation changes hence explain the variability in the spatial pattern of $\Delta\delta^{13}C_{DIC}$ in Fig. 9A. In particular, the deepening and strengthening of the AMOC during

the deglaciation lead to positive $\Delta\delta^{13}C_{DIC}$ changes in the deep Atlantic and to negative $\Delta\delta^{13}C_{DIC}$ changes in the upper Atlantic, consistent with proxy reconstructions. The small mean change in $\Delta\delta^{13}C_{DIC}$ results from a complex interplay of processes affecting the export of POC and $CaCO_3$ as well as sedimentation-weathering imbalances.

The sensitivity simulations may be grouped by processes with small and large changes in the spatial pattern of $\Delta\delta^{13}C_{DIC}$. $\Delta$land, coral reef regrowth, and organic weathering flux affect the input of carbon and carbon isotopes into the atmosphere–

ocean system, leading to relatively uniform patterns of change (Fig. 9B, F, and H). Changes in SO wind stress, SO gas transfer rate, rain ratio, and upper ocean remineralization profile on the other hand mainly lead to a redistribution of carbon and carbon isotopes in the atmosphere–ocean system amplified by sediment interactions (Fig. 9C,D,E,G).

The removal of light land carbon leaves the atmosphere and ocean relatively enriched in $\delta^{13}C$ (Fig. 9B). As a result of the carbon removal from the coupled system, changes in bottom water concentrations of $CO_3^{2-}$ and $O_2$ may influence sedimentation



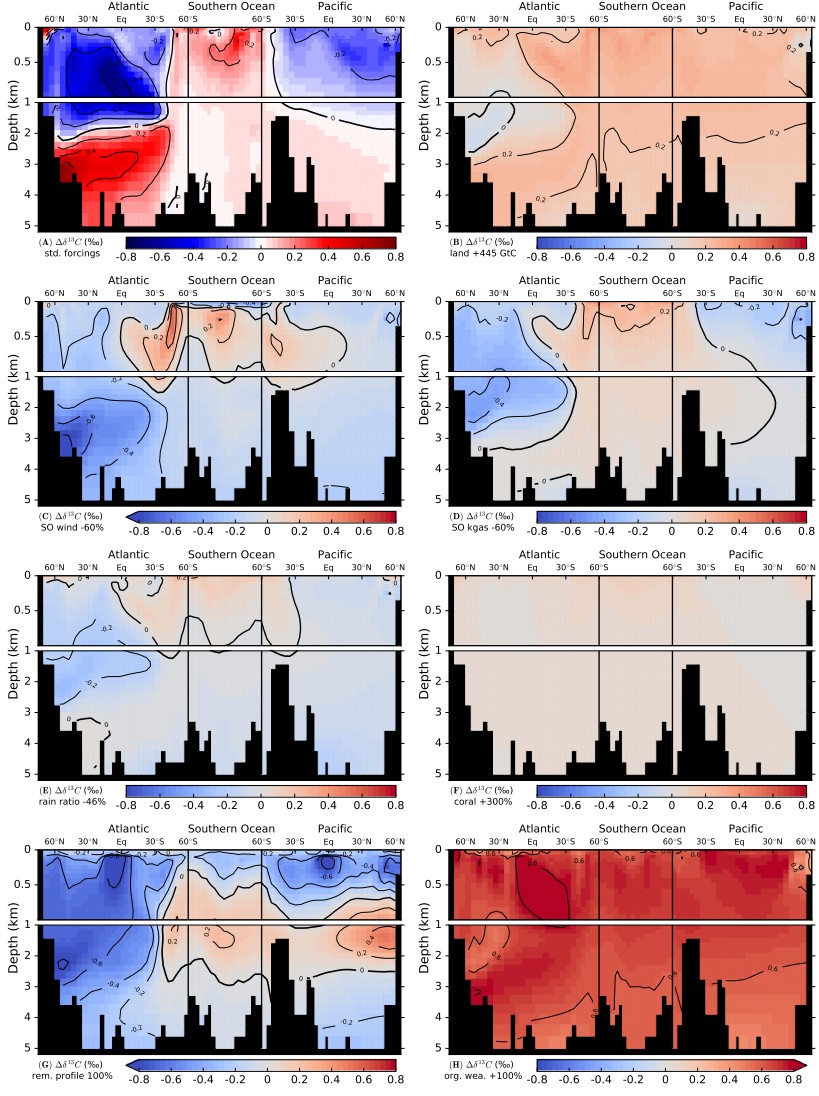

**Figure 9.** Simulated deglacial change in the isotopic signature of DIC, $\Delta\delta^{13}C_{DIC}$ (PI minus LGM). Values are displayed along sections across the Atlantic, Southern Ocean, and Pacific and in response to standard deglacial forcings and (B) to (H) due to changes in individual mechanisms as inferred from factorial simulations. $\Delta\delta^{13}C_{DIC}$ are for a deglacial carbon uptake of 445 GtC by the land biosphere (B), Southern Ocean wind stress (C) and gas transfer rate (D) reduced by 60 % at LGM relative to PI, rain ratio as reduced by -45 % at LGM relative to PI (E), an additional deglacial shallow water carbonate deposition of ∼1,500 GtC (F), a change in the organic matter remineralization between a linear profile and the standard depth-scaling (G), and for a 100 % increase in the organic weathering flux (H). The absolute change in $\Delta\delta^{13}C_{DIC}$ is shown in panel (A) for the standard forcings (bright colors), while panel (B) to (H) show the difference relative to the standard run (pastel-colored).





fluxes of POC and CaCO$_3$ locally, and lead to small local patterns (see for instance negative anomalies in $\Delta\delta^{13}C_{DIC}$ in the NADW region of Fig. 9B). Generally though, a smooth pattern emerges for changes in $\Delta$land. Similarly, the doubling of the organic weathering flux during the glacial results in a strong uniform change in $\Delta\delta^{13}C_{DIC}$ (Fig. 9H and Fig. 7A) as more isotopically light carbon is added to the system at the LGM than PI. Increased rates of coral reef regrowth have little impact

on $\Delta\delta^{13}C_{DIC}$ as the isotopic signature is similar for DIC and deposited CaCO$_3$ (Fig. 9F). The slight increase in $\Delta\delta^{13}C_{DIC}$ in response to enhanced coral reef regrowth has been attributed to changes in the fractionation during marine photosynthesis (Freeman and Hayes, 1992; Menviel and Joos, 2012).

Changes in the other four mechanisms not only affect the mean $\Delta\delta^{13}C_{DIC}$ (see Fig. 7A), but also change the spatial pattern significantly (Fig. 9C, D, E, and G). As described in detail by Tschumi et al. (2011) and Menviel et al. (2015), a more poorly

ventilated ocean due to reduced wind stress leads to an increase in $\delta^{13}C_{atm}$ and a subtle initial decrease in $\delta^{13}C_{DIC}$. Changes in temperature and circulation in response to decreased Southern Ocean wind stress lead to lower marine oxygen concentrations and thus an increase in POC sedimentation. The following removal of $^{13}$C depleted carbon from the ocean increases $\delta^{13}C_{DIC}$. Restoring wind stress to its initial value over the deglacial reverses the changes described above. As the sediment feedbacks act on longer timescales, propagation of the signal to the whole ocean is not yet achieved (Fig. 9C). A complex interplay of

processes such as changes in circulation and subsequent changes in export fluxes, oxygen concentrations, and remineralization of POC as well as changes in the lysocline and CaCO$_3$ cycling spatially overlay each other, yielding the $\Delta\delta^{13}C_{DIC}$ distribution seen in Fig. 9C.

Ocean–sediment interactions also play an important role for $\delta^{13}C_{DIC}$ in the case of changes in the Southern Ocean gas transfer rate. Decreasing the gas transfer rate leads to a strong increase in $\delta^{13}C_{atm}$ and an initial subtle decrease in $\delta^{13}C_{DIC}$.

The ocean–sediment interactions are very similar to as described for SO wind stress changes above. The sedimentation fluxes of POC and CaCO$_3$ increase in response to reduced oxygen concentrations from a lower gas transfer rate and changes in the CO$_3^{2-}$ concentration, as does the $\delta^{13}$C signature of the flux, leading to a removal of $^{13}$C from the ocean. From LGM to PI the opposite process takes places (increasing gas transfer back to its initial value), however, due to the long timescales of ocean–sediment interactions, the signal has not propagated to the whole ocean and in the Atlantic and Pacific negative $\Delta\delta^{13}C_{DIC}$

from the glacial still prevail (Fig. 9D).

As discussed in detail in Tschumi et al. (2011), responses in $\delta^{13}$C from reducing the export rain ratio are small in both the ocean and atmosphere (Fig. 7A-B) and arise from weathering-sedimentation imbalances and changes in the isotopic value of CaCO$_3$ and POC. This negative perturbation in $\delta^{13}$C is not removed entirely over the deglacial and Holocene due to the long ocean–sediment response timescale, leaving the generally negative $\Delta\delta^{13}C_{DIC}$ as seen in Fig. 9E.

Finally, a change to a linear remineralization profile in the upper 2 km leads to a dipole pattern in $\delta^{13}C_{DIC}$ with enriched values in the upper 1 km of the ocean and depleted values underneath (not shown). Over the deglacial, the process is reversed, yielding the pattern in $\Delta\delta^{13}C_{DIC}$ seen in Fig. 9G from the overlap of different response timescales. Overall, applied changes in the remineralization profile lead to loss of carbon from the atmosphere–ocean system from LGM to PI.

In summary, the carbon and carbon isotope balance in our model simulations can change as a result of altered input fluxes,

from changes in the surface ocean $\delta^{13}$C signatures which are then reflected in POC and CaCO$_3$ export fluxes and coral reef



regrowth, and from altered sedimentation fluxes as a result of altered bottom water $CO_3^{2-}$ and $O_2$ concentrations. For all seven mechanisms, weathering-sedimentation imbalances and feedbacks exert an impact on the carbon and carbon isotope budget and need to be taken into account on glacial–interglacial timescales. Also, the overlap of different response timescales for atmosphere–ocean and ocean–sediment interactions shape the patterns in $\Delta\delta^{13}C_{DIC}$ shown in Fig. 9.

## 4   Discussion

We show that carbon exchange with ocean sediments and the lithosphere biases earlier $\delta^{13}$C-based estimates of deglacial change in land carbon storage significantly low. This finding appears very robust considering modern estimates and reconstructions of ocean– sediment and weathering fluxes, and our understanding of the marine carbonate chemistry. These geological
fluxes are an integral part of the Earth system. Many earlier estimates of land carbon storage change rely on the budget of the stable carbon isotope, but neglected the isotopic exchange fluxes with marine sediments and the lithosphere. This approach became popular, perhaps as it is simple and transparent to solve two global budget equations. However, results from idealized pulse-release experiment as well as from transient deglacial simulations show that this neglect is not justified. The importance of ocean sediment and lithosphere fluxes in regulating past atmospheric $CO_2$ and carbon, isotopes, nutrient, and alkalinity
inventories in the ocean is also emphasized in previous modelling studies (e.g. Huybers and Langmuir, 2009; Tschumi et al., 2011; Roth and Joos, 2012; Roth et al., 2014; Wallmann et al., 2015; Heinze et al., 2016). Data-based reconstructions of burial fluxes of calcium carbonate and organic matter (Cartapanis et al., 2016, 2018) reveal that these fluxes are more dynamic than often assumed.

For cost-efficiency, we build a reduced form emulator of the Bern3D model explicitly simulating the carbon exchange
with ocean sediments and the lithosphere. Orbital changes and radiative forcing by greenhouse gases and ice albedo as well as freshwater input fluxes are taken into account. The strength of seven generic, archetypical deglacial carbon cycle processes was varied in a Bayesian Monte Carlo approach using 500,000 realizations. The model outcomes are used to constrain the deglacial change in the land biosphere inventory using four observational constraints. These constraints are the deglacial changes in marine ($\Delta\delta^{13}C_{DIC}$) and atmospheric ($\Delta\delta^{13}C_{atm}$) carbon isotopic composition, in atmospheric $CO_2$ ($\Delta CO_2$), and in deep
Pacific carbonate ion concentration ($\Delta CO_3^{2-}$). This yields an observation-constrained probability distribution for the change in land carbon inventory, $\Delta$land, between the preindustrial (PI) and Last Glacial Maximum (LGM).

### 4.1   Uncertainties of this study

There are a number of uncertainties that are not explicitly addressed by our approach. These may affect the median and distribution of our reconstruction of the deglacial change in land carbon inventory ($\Delta$land). We rely on characteristic forcing–
response relationship as obtained with the Bern3D model and these may be different than in reality. Shortcomings related to coarse model resolution, simplified ocean dynamics, global parameterizations of ocean–sediment interactions or highly parameterized representation of biological processes may bias model outcome. Further, the current crop of models, including

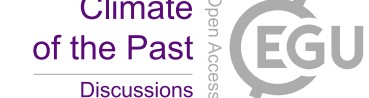



the Bern3D model, is not able to freely simulate reconstructed variations in atmospheric $CO_2$ and other biogeochemical variables over the deglacial period. Nevertheless, the Bern3D model simulates the modern distribution of a range of water mass, ventilation, and biogeochemical tracers in good agreement with observational data (Roth et al., 2014). Global inventories and spatial distribution of biogenic $CaCO_3$, organic carbon, and opal in the sediments as well as their fluxes between the ocean,

marine sediments and the lithosphere are in agreement with pre-industrial data-based estimates (see Appendix B). The Bern3D model simulates a weaker and shallower Atlantic Meridional Overturning Circulation at the LGM compared to the PI and the simulated anomalies in $\delta^{13}C$ of DIC agree with corresponding reconstructions (see Fig. 9 in Menviel et al., 2012). Regarding dissolved oxygen, qualitative reconstructions show a general decrease in oxygenation at intermediate depths and increase in the deep ocean (Jaccard and Galbraith, 2012; Jaccard et al., 2014; Galbraith and Jaccard, 2015) across the deglaciation. This

pattern is also reproduced (not shown) in Bern3D model simulations that match/come close to matching the four data-based constraints.

We selected a set of seven generic, archetypical processes that are varied in deglacial simulations in addition to the processes explicitly implemented in the Bern3D model. These archetypical processes are intended to represent the space of multi-proxy response relationships (Fig. 8) for all the different processes which plausibly influenced deglacial carbon cycle changes. This is

a simplification. Though, we argue that the seven selected processes roughly cover the plausible range of multi-proxy responses (Fig. 8). In addition, uncertainties in the importance of these processes are explicitly considered in our Monte Carlo approach and contribute to the uncertainty range in our estimate of $\Delta$land.

For example, there are various carbon reservoirs with an isotopic light signature very similar to that of plant and soil carbon. These include organic carbon stored in ocean sediments and shelves (Broecker, 1982; Ushie and Matsumoto, 2012; Cartapanis

et al., 2016), dissolved organic carbon (DOC) in the ocean (Hansell, 2013), or gaseous $CO_2$ stored in the unsaturated zones of aquifers from the decomposition of organic material (Baldini et al., 2018). Any carbon release or uptake from these reservoirs might be wrongly attributed to a change in the carbon inventory of the land biosphere given their very similar $\delta^{13}C$ signature and similar multi-proxy response relationships (e.g., compare pink and black line in Fig. 8A-D for changes in organic weathering and land carbon). Indeed, changes in these reservoirs have been invoked as an alternative explanation for the reconstructed

deglacial change in marine $\delta^{13}C$ (Broecker, 1982; Zimov et al., 2006, 2009; Zech et al., 2011; Kemppinen et al., 2018). Here, we represent variations in these carbon stocks by varying weathering-burial imbalances of organic material in addition to adding variable amounts of isotopically light land carbon to the atmosphere in our 500,000 member ensemble. Thereby, we consider uncertainties related to inventory changes in these other isotopically light reservoirs as further elaborated in the next paragraph.

Information in Baldini et al. (2018) (their Table 1) implies that the change in $CO_2$ stored in the unsaturated zone of aquifers, neglected in the Bern3D model, is in the range of 12 to 86 $GtC$ between the Last Glacial Maximum and the preindustrial period. This range is small compared to the estimated change in land carbon of around 850 $GtC$. Perhaps more important, the modern inventory of refractory DOC is about 650 $GtC$ (Hansell, 2013) and assumed constant in Bern3D, while changes in the small inventory of labile DOC are explicitly simulated. It is unclear how the DOC pool varied over the past. Lower temperature

might have favored the preservation of DOC, while a smaller vegetation cover on land might have reduced the input of DOC





into the ocean. Cartapanis et al. (2016) suggest an approximately linear decrease in the flux of organic carbon transferred to deep ocean sediments from around 28 GtC kyr$^{-1}$ at the LGM to around 17 GtC kyr$^{-1}$ during the Holocene, corresponding to a cumulative imbalance of around 55 GtC. The change in organic carbon stocks stored in sediments on shelves is uncertain and may amount to several 100s of GtC over the deglaciation. Here, we varied changes in organic carbon sources to the atmosphere

ocean system over a very wide range in comparison to the inventory changes discussed above. Organic-like carbon input by weathering at the LGM is varied in the unconstrained ensemble within 50% to 200% of its modern value. This corresponds to a cumulative PI-LGM anomaly in this input flux of +780 to -1,560 GtC. These anomalies in input, though not directly comparable to inventory changes, are large in comparison to the inventory changes discussed here. In summary, a potential misattribution of changes in other organic carbon reservoirs to changes in $\Delta$land is included in our uncertainty estimate of

$\Delta$land.

Uncertainties related to changes in the $\delta^{13}$C signature of land carbon, also leading to a potential misattribution when estimating $\Delta$land, are thought to be small on the global scale (Joos et al., 2004; Ciais et al., 2012). The $\delta^{13}$C signature of the terrestrial carbon uptake/release in this study was set to -24 ‰. In past studies, also slightly different $\delta^{13}$C signatures for the terrestrial biosphere have been used, ranging from about -25 ‰ to -23 ‰. To test the robustness of our $\Delta$land estimate to the

inferred $\delta^{13}$C signature of the terrestrial carbon, two series of experiments were conducted with $\delta^{13}$C signatures of -25 ‰ and -23 ‰. The new sensitivities of the data-based target constraints to changes in $\Delta$land were determined and 100,000 parameter combinations were calculated. Resulting estimates of $\Delta$land vary only by a few tens of GtC for both $\delta^{13}$C signatures. Considering the spread of possible $\Delta$land estimates in this study, the exact value of the $\delta^{13}$C signature of the terrestrial uptake within the considered range thus seems to play a minor role in determining $\Delta$land.

Our approach is limited in that we only consider the change between LGM and PI for four relevant proxy variables. This restriction allowed us to build a cost-efficient and non-linear emulator to explore a very large range of parameter combinations. Future efforts to refine estimates of $\Delta$land may explicitly consider the temporal evolution of the different proxies (Schmitt et al., 2012; Yu et al., 2014; Eggleston et al., 2016; Bauska et al., 2016; Peterson and Lisiecki, 2018) to study biogeochemical changes over the past glacial–interglacial cycles (Menviel et al., 2012; Heinze et al., 2016; Ganopolski and Brovkin, 2017). Heinze et al.

(2016) applied a linear inverse modelling approach to approximate the evolution of about eighty proxy records for atmospheric $CO_2$ and sedimentary $CaCO_3$, opal, and $\delta^{13}$C of benthic and planktonic foraminifera over the last 130 kyr. This was done by scaling the magnitude of seven processes with the ice core deuterium signal in their annual mean circulation–biogeochemical model. Here, we explicitly consider seasonality and non-linearities in forcing–response relationships and restrict the focus to the deglacial period. Additional proxy data may be included in future work. These include, for example, changes in oxygen

(e.g. Jaccard and Galbraith, 2012; Galbraith and Jaccard, 2015; Hoogakker et al., 2015; Gottschalk et al., 2016), in export productivity and burial (e.g. Kohfeld et al., 2005; Anderson and Burckle, 2009; Cartapanis et al., 2016), in radiocarbon (e.g. Skinner et al., 2017), or in isotopes of protactinium, thorium, and neodymium (e.g. Böhm et al., 2015). Another avenue for improvement is to consider spatial patterns such as the documented changes in the patterns of $\delta^{13}$C of DIC between the upper and deep, and the Atlantic and Indo-Pacific ocean (e.g. Oliver et al., 2010; Peterson and Lisiecki, 2018) and in the pattern of

$CO_3^{2-}$ (Fig 6 in Menviel et al., 2012; Yu et al., 2014, and references therein).



## 4.2 Contribution of deglacial mechanisms to the CO₂ rise

Turning to the role of different mechanisms for the deglacial increase in atmospheric $CO_2$, we find that ocean circulation changes only partly explain the deglacial $CO_2$ rise. Ocean circulation changes are not able to simultaneously explain glacial–interglacial changes in $\delta^{13}C_{DIC}$ and $CO_2$ in our model (see Fig. 8A), in agreement with earlier studies (e.g., Tschumi et al. (2011); Heinze et al. (2016).

A modest deepening of the remineralization of organic matter in the upper ocean leads to large changes in atmospheric $CO_2$, while at the same time hardly changing deep Pacific $CO_3^{2-}$. This holds also for other biological mechanisms such as an increased nutrient utilization in response to iron fertilization or an elevated phosphor input to the ocean (Menviel et al., 2012). This renders this class of processes to be potentially important to explain part of the deglacial $CO_2$ increase, as reconstructions suggest small LGM to PI changes in deep Pacific $CO_3^{2-}$ concentrations (Yu et al., 2014). Accordingly, in all Monte Carlo realizations the remineralization depth of organic matter is deeper at the LGM than PI in our approach. We note that the modification in the remineralization profile of particulate organic matter is different than those applied by Menviel et al. (2012) and Roth et al. (2014) in simulations with the Bern3D model. Here, only the remineralization profile in the upper 2,000 m is changed and, in contrast to these earlier studies, it is assumed that the fraction of exported particles that reach the sediments below 2,000 m is not altered. The implicit assumption is that cooler temperatures at the LGM than PI only affect the dissolution of relatively small or labile particles in the upper ocean in a significant way, while large or refractory particles sink fast enough to reach ocean sediments both under LGM and PI conditions.

Our solutions also point to a significant role of alkalinity based mechanisms for the deglacial $CO_2$ increase. A decrease in the surface ocean alkalinity tends to increase the $CO_2$ partial pressure and in turn atmospheric $CO_2$. Surface ocean alkalinity is governed by the cycling of $CaCO_3$ and, to a smaller extent, of organic carbon within the ocean as well as by the balance of alkalinity input by weathering and alkalinity loss by burial of $CaCO_3$, and, to a smaller extent, by organic material. The constrained ensemble yields a large extra burial of $CaCO_3$ as well as an increase in the rain ratio of $CaCO_3$ to organic matter by particle export over the deglaciation. This is qualitatively consistent with the reconstruction of Cartapanis et al. (2018) who suggest that the $CaCO_3$ burial flux (below 200 m) increased by about 0.2 $\mathrm{GtC\,yr^{-1}}$ from the LGM to the Holocene. This suggests that changes in alkalinity based mechanisms contributed to the deglacial $CO_2$ increase. In summary, our multi-proxy approach suggests, in agreement with earlier studies (e.g. Tschumi et al., 2011; Menviel et al., 2012; Heinze et al., 2016; Ganopolski and Brovkin, 2017; Shaffer and Lambert, 2018) that a combination of processes is responsible for glacial–interglacial $CO_2$ variations, while the attribution of change to individual mechanisms remains difficult. In addition, several processes that were potentially important for the reorganisation of the deglacial carbon cycle influence the mean ocean $\delta^{13}C$ signature significantly. Their influence cannot be ignored when analyzing the budget of $\delta^{13}C$ on deglacial timescales.

## 4.3 Comparison to other estimates of Δland

Δland estimates range from -400 GtC, e.g. a larger terrestrial carbon inventory during the glacial (Zimov et al., 2006, 2009; Zech et al., 2011; Kemppinen et al., 2018), to 1,500 GtC (Adams and Faure, 1998), based on a variety of methods each





associated with its own uncertainties and limitations. Studies suggesting a negative $\Delta$land (Zimov et al., 2006, 2009; Zech et al., 2011; Kemppinen et al., 2018) argue that the reconstructed deglacial $\delta^{13}C_{DIC}$ change is due to changes in other organic carbon reservoirs. These postulated changes would not only need to explain the observed deglacial change in $\delta^{13}C_{DIC}$, but in addition also explain the imprint of a negative $\Delta$land on the isotopic budget. Considering additional data-based constraints, as

proposed in this study, suggests negative $\Delta$land values to be highly unlikely. In addition, our results show that $\Delta$land is likely larger than 450 GtC. Otherwise, solutions for $\delta^{13}C_{DIC}$ are significantly biased low compared to the data-based reconstruction.

On the other hand, $\Delta$land estimates at the very high end are also in conflict with some data-based constraints and Hoogakker et al. (2016) point to the difficulty of deriving $\Delta$land estimates based on pollen data. We will illustrate some of these issues with the example of the study by Adams and Faure (1998) that provides reconstructions of $\Delta$land for three time slices, 18 kyrBP,

8 kyrBP, and 5 kyrBP and assumes that about 1,400 GtC of the regrowth of the terrestrial biosphere occured between 18 and 8 kyrBP. This is in conflict with the evolution of the terrestrial biosphere uptake as proposed by Elsig et al. (2009). In their best guess scenario Elsig et al. (2009) assume a growth of the terrestrial biosphere of about 150 GtC between 11 and 8 kyrBP and 700 GtC between 18 and 11 kyrBP which is only 60% of the amount given by Adams and Faure (1998). While the growth of the terrestrial biosphere from 18 to 11 kyrBP is a scenario estimate, the growth over the Holocene is constrained

based on $\delta^{13}C_{atm}$ measurements to $\sim$250 GtC (Elsig et al., 2009). Assuming the total estimate of 1,500 GtC was right, this would leave more than 1,200 GtC to be transferred to the terrestrial biosphere between the LGM and the beginning of the Holocene. Considering possible processes that could achieve such a large transfer seems difficult. Generally this might pose an upper limit to $\Delta$land.

Our estimate of $\Delta$land appears consistent with a recent biome-based reconstruction of soil carbon storage (Lindgren et al.,

2018). Lindgren et al. (2018) report an increase in soil carbon inventory of about 400 GtC for the region north of about 30°N and considering changes in permafrost, mineral and peatland as well as in loess and subglacial soil. This is in agreement with our $\Delta$land estimate, when assuming an increase in carbon stored in vegetation and in tropical and southern hemisphere ecosystems and soils. Furthermore, modeling studies have generally yielded positive $\Delta$land estimates of about 400–900 GtC (Joos et al., 2004; Prentice et al., 2011; Brovkin et al., 2012; O'Ishi and Abe-Ouchi, 2013; Ganopolski and Brovkin, 2017;

Davies-Barnard et al., 2017).

In summary, many lines of evidence point to a positive $\Delta$land and we suggest a likely range of about 450 to 1,250 GtC. Yet, the uncertainty in this estimate remains large, owing to both uncertainties in processes and in observational constraints. Unfortunately, the necessity to include exchange with sediments and the lithosphere and other deglacial processes in the isotopic budget adds complexity. This tends to increase the uncertainty range compared to the simplified and biased assessments that

consider the isotopic budget in the closed ocean–land–atmosphere system. Further research is necessary to better understand the temporal evolution of the glacial/interglacial carbon cycle.



## 5   Conclusions

We used a Bayesian approach to constrain the change in the land biosphere carbon inventory between the Last Glacial Maximum and the preindustrial period. Four data-based constraints are applied: the deglacial change in the $\delta^{13}$C signature of atmospheric $CO_2$ and of dissolved inorganic carbon in the ocean, the deglacial change in atmospheric $CO_2$ and in deep Pacific
carbonate ion concentrations. The strength of generic, archetypical mechanisms for deglacial biogechemical changes was varied in 500,000 simulations in an emulator, built from a large suite of Bern3D model simulations. Carbon, nutrient, alkalinity, and carbon isotope exchange with the lithosphere and ocean sediments is explicitly taken into account, in contrast to earlier studies which applied the isotopic budget of $^{13}$C to estimate deglacial change in land biosphere carbon.

We demonstrate in idealized and transient deglacial simulations that isotopic exchange with ocean sediments and the litho-
sphere is important for the budget of $^{13}$C on timescales of the deglaciation. We find that the carbon stocks in the land biosphere were likely around 450 to 1,250 GtC ($1\sigma$ range) larger at preindustrial times than at the last glacial maximum. The median estimate is $\sim$850 GtC. This is much larger than earlier estimates based on the budget of the stable carbon isotope $^{13}$C. These earlier studies neglected interactions with the lithosphere and ocean sediments and the influence of other deglacial carbon cycle processes on the isotopic budget. This neglect biases their estimate significantly low. Our multi-proxy approach suggests that a
combination of different mechanisms contributed to the deglacial increase in $CO_2$. These include, aside from ocean circulation, temperature, and salinity changes, a net removal of alkalinity from the surface ocean by increased burial of calcium carbonate in coral reefs and ocean sediments and potentially increasing export of calcium carbonate from the surface to the deep ocean. Further, changes in biological mechanisms such as a deglacial shoaling of the remineralization depth for organic matter may have been instrumental to increase atmospheric $CO_2$. The results demonstrate that ocean sediments and the weathering–burial
cycle are an integral part of the Earth system playing a fundamental role on glacial– interglacial timescales.

*Data availability.*   Data used for this study are available upon request to the corresponding author (jeltsch@climate.unibe.ch).

## Appendix A: The Bern3D model and pulse experiments

### A1   The Bern3D model

The Bern3D EMIC features a three dimensional geostrophic ocean (Müller et al., 2006; Edwards et al., 1998) with an isopycnal
diffusion scheme and Gent–Mc Williams parameterization for eddy-induced transport (Griffies, 1998). The model includes a thermodynamic sea-ice component coupled to a single layer energy–moisture balance atmosphere (Ritz et al., 2011). Further, a sediment module (Tschumi et al., 2011; Heinze et al., 1999), and a 4-box terrestrial biosphere (Siegenthaler and Oeschger, 1987) are coupled to the model.

The horizontal resolution is 41x40 grid cells (Roth et al., 2014; Battaglia and Joos, 2018a, b) and is the same for the ocean,
atmosphere, and sea-ice components. In the vertical, the ocean has 32 logarithmically spaced layers. Wind stress at the surface



is prescribed following the NCEP/NCAR monthly wind stress climatology (Kalnay et al., 1996). Carbonate chemistry, air–sea gas exchange for $CO_2$ and $^{14}CO_2$ is implemented according to OCMIP-2 protocols (Najjar and Orr, 1999; Orr et al., 1999) with updates for the $^{14}C$ standard ratio and half-life (Orr et al., 2017), the calculation of the Schmidt number (Wanninkhof, 2014) and the carbonate chemistry (Orr and Epitalon, 2015). In order to match observational estimates of natural and bomb-

produced $^{14}C$, the global mean air–sea transfer is reduced by 19 % compared to OCMIP-2 in order to match observation-based radiocarbon estimates (Müller et al., 2008). Another update with respect to the OCMIP-2 protocols concerns the gas transfer velocity which scales linearly with wind speed following Krakauer et al. (2006). Analog and consistent formulations are used for air–sea exchange of oxygen and $^{13}CO_2$.

Biogeochemical cycling in the model is detailed in Tschumi et al. (2011) and Parekh et al. (2008) with further documen-
tation of results in follow-up studies (e.g. Menviel et al., 2012; Menviel and Joos, 2012; Roth and Joos, 2012; Roth et al., 2014; Menviel et al., 2015; Battaglia et al., 2016; Battaglia and Joos, 2018b, a). Dissolved inorganic carbon and semi-labile organic carbon (DIC, DOC), the corresponding isotopic forms ($DI^{13}C$, $DO^{13}C$, $DI^{14}C$, $DO^{14}C$), as well as alkalinity (Alk), phosphate ($PO_4$), oxygen ($O_2$), iron (Fe), silica (Si), and an ideal age tracer are explicitly transported by advection, diffusion, and convection. New production of organic matter is limited to the euphotic zone in the uppermost 75 m and calculated as
a function of light availability, temperature, and phosphate and iron availability (Doney et al., 2006). Two thirds of the new production are transferred to the pool of dissolved organic carbon (DOC) and the remainder is exported as particulate organic carbon (POC) (Tschumi et al., 2008). The biological fluxes of C, $PO_4$, Alk, and $O_2$ and the corresponding elemental ratios in DOC and POC are linked by fixed Redfield ratios (P:Alk:C:$O_2$:Alk=1:17:117:-170; Sarmiento and Gruber, 2006). DOC decays with a life time of 0.5 yr. POC is remineralized following a Martin's Curve (Martin et al., 1987) given by

$$F_{POC}(z) = F_{POC}(z_0) \times \left(\frac{z}{z_0}\right)^{-\alpha} \ for \ z > z_0 \tag{A1}$$

where $z_0$ refers to the reference depth of 75 m (see also Roth et al., 2014). Export of calcium carbonate ($CaCO_3$) and of opal is computed from new production and availability of dissolved silica in the euphotic zone. The export rain ratio of $CaCO_3$ to POC is set to 0.083 in the absence of dissolved silica, while the production ratio of $CaCO_3$ to POC is reduced at the favor of opal production in regions with abundant availability of silicic acid. Particulate $CaCO_3$ and opal is remineralized below the
euphotic zone with an e-folding depth scale of 5,066 m and 10,000 m, respectively.

A 10-layer sediment diagenesis module (Tschumi et al., 2011; Heinze et al., 1999) is used to explicitly calculate transfer fluxes to and redissolution fluxes from reactive sediment to the water column as well as loss fluxes to the lithosphere for nutrients, carbon and carbon isotopes. In equilibrium, loss fluxes to the litosphere are equal to input fluxes by weathering. The sediment module dynamically calculates the transport, redissolution/remineralization, and bioturbation of solid material, the
porewater chemistry, and diffusion in the top 10 cm of the sediment. Four solid tracers ($CaCO_3$, opal, POC, clay) and seven tracers in the porewater (DIC, $DI^{13}C$, $DI^{14}C$, alkalinity, phosphate, oxygen, and silicic acid) are modelled. The dissolution of $CaCO_3$ depends on the pore-water $CO_3^{2-}$ concentration. The oxidation rate of POC, on the other hand, depends on oxygen concentrations in the porewater and the weight fraction of POC in the solid phase of the sediment. Denitrification is not considered in this model version, but $O_2$ is not consumed below a threshold, somewhat reflecting the process of denitrification



without modeling $NO_3^-$. The respective reaction rate parameters for $CaCO_3$ dissolution and POC oxidation are global constants (see Roth et al., 2014). The model assumes conservation of volume, i.e. the entire column of the sediments is pushed downwards if deposition exceeds redissolution into pore waters. Any solid material that is pushed out of the diagenetic zone (top 10 cm) disappears into the subjacent diagenetically consolidated zone (burial or loss flux) (see Tschumi et al., 2011, for more details).

Carbonate chemistry within sediment pore waters is calculated as in the ocean by using the MOCSY routine of Orr and Epitalon (2015).

Weathering fluxes of $PO_4$, Alk, DIC, $DI^{13}C$, and silic acid are added uniformly to the coastal surface ocean. The global weathering fluxes are set equal to the burial fluxes diagnosed at the end of the model spin-up. In the standard model setup these diagnosed input fluxes are kept constant. The pulse experiments include a setting where the input fluxes vary as a function of

global mean surface air temperature and $CO_2$ following Colbourn et al. (2013). One of the factorial sensitivity runs assesses the impact of alterd input fluxes of organic material across the deglacial. This weathering–burial cycle and the associated burial–nutrient feedbacks (Tschumi et al., 2011) are important for the mass balance of $^{13}C$ and for the removal of marine perturbations in nutrients, carbon, and isotopes on deglacial time-scales. In other words, these processes directly affect the estimate of the change in land carbon from the $^{13}C$ mass balance.

The atmosphere exchanges carbon and carbon isotopes with a four box land biosphere model (Siegenthaler and Oeschger, 1987). It includes a mass of 2,220 GtC and represents the carbon that is actively exchanged with the atmosphere. This 4-box reservoir is only used to simulate the dilution of an atmospheric isotopic perturbation by the land biosphere, but not to address changes in land carbon stocks. The carbon inventory of the land biosphere, and thus the dilution of an isotopic perturbation, is about 20 times smaller than that of the ocean–atmosphere–sediment system.

$^{13}C$ stocks and flows are modeled in the atmosphere–ocean–sediment–landbiosphere system. $^{13}C$ fluxes to the lithosphere associated with the burial of POC and $CaCO_3$ and weathering input are explicitly simulated. $^{13}C$ fractionation is considered for air–sea gas transfer, for carbonate chemistry, for the formation of $CaCO_3$, POC, DOC, and for photosynthesis on land, while no fractionation is assumed during remineralization on land and in the ocean. Gross air–to–sea and gross sea–to–air fluxes of $^{13}CO_2$ are implemented considering kinetic fractionation (Siegenthaler and Oeschger, 1987) and strongly temperature–

dependent equilibrium fractionation between the various carbonate species (Mook, 1986). Isotopic fractionation during $CaCO_3$ formation is small and computed following Mook (1986). This results in an isotopically heavy signature of around 2.9 ‰ for $CaCO_3$. Fractionation during photosynthesis in the ocean and thus between dissolved $CO_2$ ($[CO_2]$) and POC and DOC is calculated according to Freeman and Hayes (1992). The fractionation increases logarithmically with $[CO_2]$ in surface water and results in an isotopically light signature for POC and DOC of around -20 ‰. A lowering of $[CO_2]$, or correspondingly of

$pCO_2$, by about 10 % yields a change in the isotopic signature of POC by +0.55 ‰. The burial of POC and $CaCO_3$ results in an isotopic signature of the burial flux of around -9 ‰, intermediate between the isotopically light POC and the isotopically heavy $CaCO_3$ signature. In equilibrium, the burial flux is compensated by a weathering input flux of equal amount and signature. On land, a constant fractionation of 18.1 ‰ is applied for simplicity.





## A2 Initialisation of Bern3D simulations

The model is spun up over 60,000 yr to a preindustrial equilibrium corresponding to 1765 CE boundary conditions similar to the approach outlined in Roth et al. (2014). $CO_2$ is set to 278 ppm and $\delta^{13}C$ of $CO_2$ to -6.305 ‰. The loss of tracers due to sedimentary burial is compensated during the spin-up by variable weathering fluxes in order to conserve oceanic inventories of tracers. After the system has equilibrated, the weathering fluxes are kept constant at the values diagnosed at the end of the spin-up in the standard setup.

## A3 Bern3D pulse experiments

Idealized experiments with a pulse-like carbon removal from the atmosphere are conducted in i) a closed system, ii) an open system, and iii) an open system model configuration allowing for weathering feedbacks. In i), only the atmosphere–ocean–land biosphere model components are used and all tracers are conserved within these three reservoirs. In ii), the sediment module is added to the configuration outlined above and the carbon and tracer inventories within the atmosphere–ocean–land biosphere system are allowed to vary due to weathering-burial imbalances. In iii), weathering fluxes vary in response to changes in atmospheric temperature and $CO_2$ concentrations following the parameterizations given in Colbourn et al. (2013), accounting for changes in $CaCO_3$ and $CaSiO_3$ weathering on land.

Simulations are started from the end of the PI spin-up and atmospheric $CO_2$ and $\delta^{13}C$ are computed in a prognostic way based on air–sea and air–land fluxes. In year 100 of the simulation, 100 GtC of carbon with a $\delta^{13}C$ signature of -24 ‰ are instantaneously removed from the atmosphere, mimicking an immediate regrowth of the terrestrial biosphere. After the pulse, the model is run for 100 kyr and anomalies are expressed relative to a control run. Control simulations show negligible changes in $CO_2$ and $\delta^{13}C$ as model drift is small.

The implied change in the land biosphere carbon stock based on a closed system assumption is calculated for all three model settings by solving the following equations for $\Delta M_{TER}$, i.e. $\Delta$land:

$$\Delta(M_A + M_B + M_O) = 0 \tag{A2}$$

$$\Delta(\delta_A M_A + \delta_B M_B + \delta_O M_O) = 0 \tag{A3}$$

with $M$ representing the inventory, $\delta$ the isotopic signature, $\Delta$ the respective time interval, and the subscripts $A$, $B$, and $O$ referring to the atmosphere, terrestrial biosphere, and ocean, respectively. The calculated $\Delta$land is then compared to the prescribed carbon uptake. This allows us to investigate the validity of the closed system assumption on different timescales.

## Appendix B: Sediment tuning

The aim of the sediment tuning was to improve the representation of export, deposition, and dissolution fluxes of POC, $CaCO_3$, and opal, as well as the distribution of weight fractions of these tracers in the sediment module of the Bern3D model after the change of the horizontal model resolution introduced in Roth et al. (2014). One concern was the total amount of $CaCO_3$ in the





**Table B1.** Old and new parameter values in the Bern3D model that were varied in the tuning of the sediment component.

| Variable | old value | new value |
|---|---|---|
| ratio of Ca:P in calcifiers | 0.3 | 0.333 |
| redissolution length scale of calcite | 2,900 m | 5,066 m |
| diffusion coefficient in the sediment pore-water | $8*10^{-6}$ cm$^2$ s$^{-1}$ | $5.5*10^{-6}$ cm$^2$ s$^{-1}$ |
| reactivity for calcite in the sediment | 1,000 l mol$^{-1}$yr$^{-1}$) | 800 l mol$^{-1}$yr$^{-1}$) |

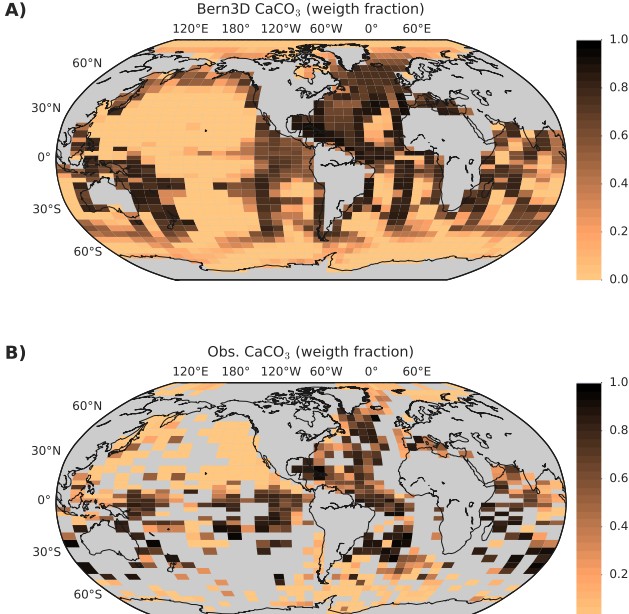

**Figure B1.** (A) Distribution of CaCO$_3$ in the Bern3D model after tuning of the sediment component under pre-industrial boundary conditions compared to (B) sediment core-top CaCO$_3$ data (Cartapanis et al., 2018). Core-top data were gridded onto the Bern3D grid and are a mean of the top 10 cm over the past 6 kyr. Grey indicates land (within coastlines) or missing data.

reactive uppermost 10 cm of the sediment represented in the sediment module. Prior to tuning, the global CaCO$_3$ content in the model was ∼400 GtC. Using observational constraints on modern global bulk mass accumulation rates (MAR) (Cartapanis et al., 2016) and global carbonate MAR (Dunne et al., 2012; Jahnke, 1996) and the density, the bulk sediment, and the porosity in the uppermost 10 cm as used in the model, we estimate a target CaCO$_3$ stock of ∼1,000 GtC. The second concern was the

5    opal burial that used to be ∼3 Tmol Si yr$^{-1}$ but with new observational data suggesting a burial flux of ∼6.5 Tmol Si yr$^{-1}$ (Tréguer and De La Rocha, 2013). Further, all other sediment related fluxes of POC, CaCO$_3$, and opal were tuned with regard to observational constraints (see Table 1).

The tuning was carried out in two steps. In the first step, three parameters were varied: the ratio of Ca:P in calcifiers, the redissolution length scale for the calcite profile, and the reactivity for calcite in the sediment. In the second step, the reactivity





of calcite and the diffusion coefficient in the sediment pore-water were varied. For both steps, a 100 member LHS ensemble was run and skill scores calculated as outlined in Steinacher et al. (2013). For the calculation of the skill scores, ocean data from GLODAP v.2 (Lauvset et al., 2016) and the World Ocean Atlas 2013 version 2 (Locarnini et al., 2013; Zweng et al., 2013; Garcia et al., 2013a, b) datasets, for export, deposition, and burial of POC, $CaCO_3$, and opal values from Battaglia et al. (2016),

5   Tréguer and De La Rocha (2013), Sarmiento and Gruber (2006), Milliman and Droxler (1996), Feely et al. (2004), and for the sediment data compilations by Cartapanis et al. (2016, 2018) were used to compare the model performance to. The distribution of $CaCO_3$ in the sediments after the tuning as compared to observational data is shown in Fig. B1. Export, deposition, and burial fluxes for POC, $CaCO_3$, and opal and the global sediment inventories in the model with respective observational ranges from the literature are given in Table 1 and the old and new parameter values in Table B1. The distribution of other tracers is

10   comparable to before and as outlined in Roth et al. (2014).

**Appendix C:  Additional Figures**





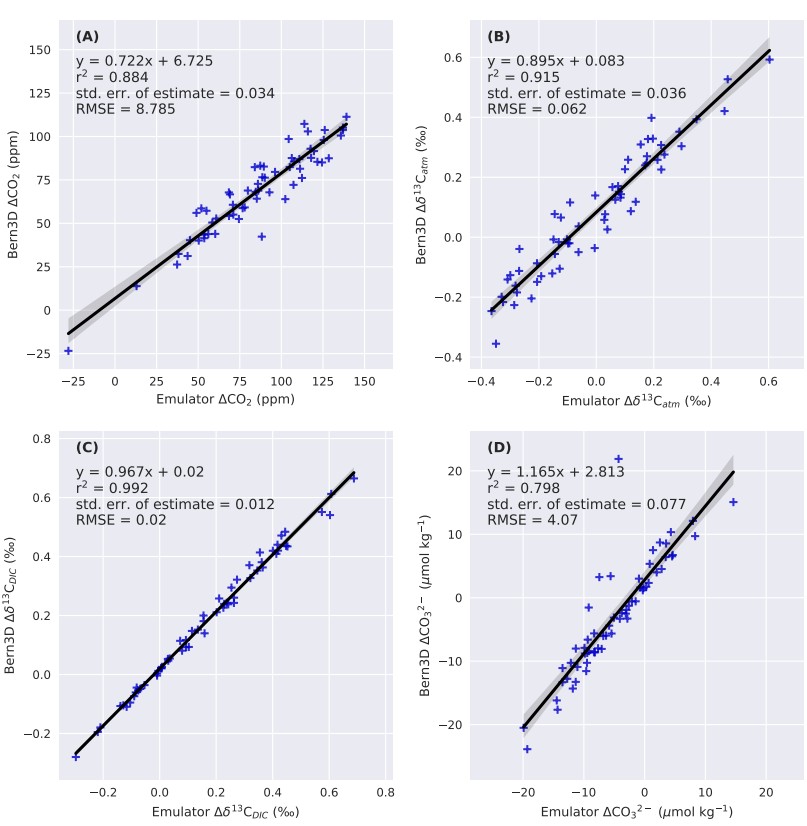

**Figure C1.** Regression of emulator results for the four data-based constraints against results from a Bern3D LHS ensemble with the same parameter combinations (as outlined in Section 2.2). The regression given in the figure is used to correct emulator results for non-linearities (Eq. 2).





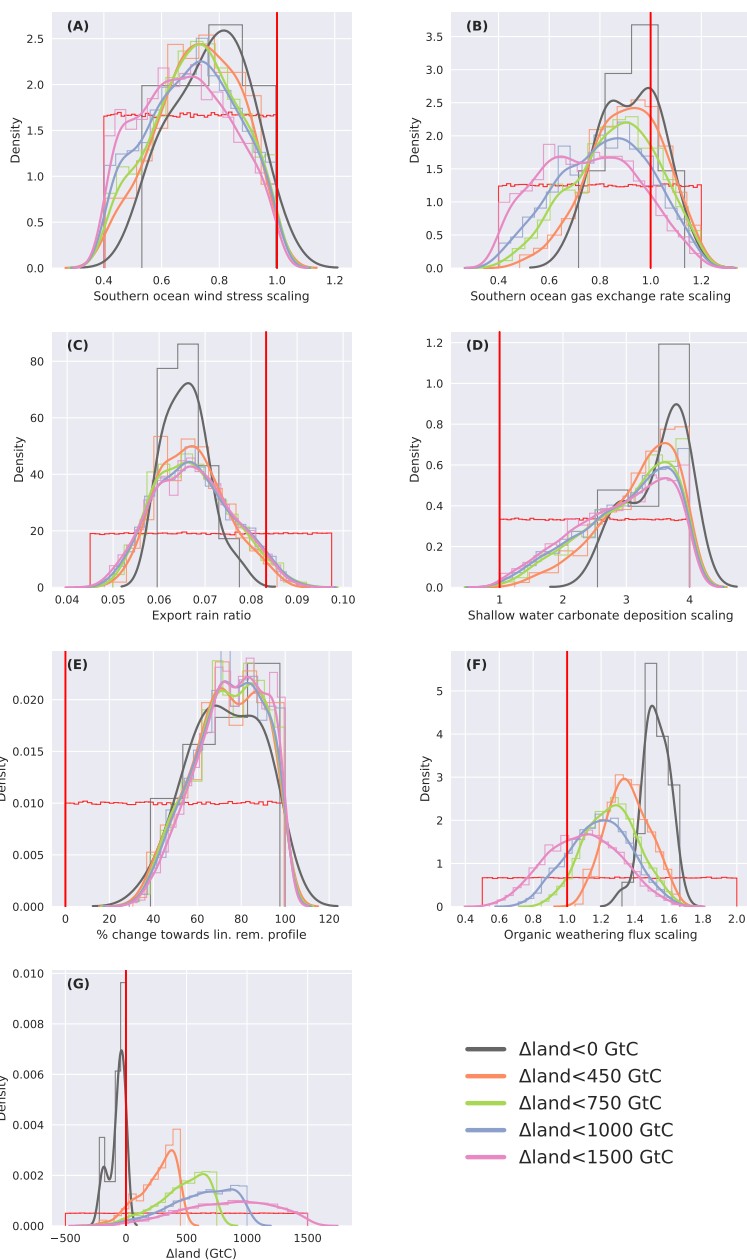

**Figure C2.** Histograms (colored thin line, normalized) and kernel density estimates (colored thick line) of applied changes in the seven mechanisms (A to G) for all emulator results that fulfill the four observational constraints. Colors of the lines correspond to sub-samples of these results according maximum changes in Δland of 1,500, 1,000, 750, 450, and 0 GtC. Thin red line gives the histogram of the prior distribution and vertical thick red line the standard parameter value. For details on the mechanisms see Section 2.2.



*Competing interests.*  The authors declare that they have no conflict of interest.

*Acknowledgements.*  A.J.-T. and F.J. acknowledge support by the Oeschger Centre for Climate Change Research and G.B. and F.J. by the
Swiss National Science Foundation (200020-172476). O.C and S.L.J were funded by the Swiss National Science Foundation (grants PP00P2-
144811 and PP00P2-172915).



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
