# Peer review of "A large increase in the carbon inventory of the land biosphere since the Last Glacial Maximum: constraints from multi-proxy data"

_Climate of the Past, 2018_

## Referee Comment (RC1) · Anonymous Referee #1 · 10 Feb 2019

Review of Jeltsch-Thommes et al. ,

Jeltsch-Thommes et al., provide a new estimate of changes in land carbon across the deglaciation. This new estimate is obtained by finding the best fit between paleoproxy constraints and results of factorial simulations with the Bern3D Earth system model coupled to Monte Carlo ensemble.

Their results suggest that there was less land carbon at the LGM than during the Holocene. Their new estimate of deglacial change in land carbon is higher than inferred by recent studies due to the inclusion of sedimentary processes.

I recommend publication in Climate of the Past, granted the comments below are taken into account.

1) The estimate of land carbon change across the deglaciation given here is higher than previous ones due to sedimentary processes. Since land carbon and organic matter have similar d13C signatures, a deglacial increase in land carbon can be partly compensated by a decrease in organic carbon burial.
It is true that in a pure mass balance approach these sedimentary might not be taken into account (even if after ~10k, these effects are not expressed fully, cf. Fig. 4). However, the model used here (Bern3D) includes a sediment model. Therefore changes in temperature, circulation, remineralization profile, etc… lead to changes in organic and carbonate burial. Sedimentary processes are therefore already taken into account in this modelling framework. I am thus confused by the inclusion of an additional "organic weathering flux". I can indeed imagine that an input of depleted d13C as would happen with a 100% increase in organic matter flux through river could significantly impact oceanic d13C. Maybe the rationale and numbers associated with the organic weathering flux should be explained in more details.

2) As a follow up on the previous comment, large parameters studies like this one are very useful to test the range of possibilities and derive statistics on possible parameter space. However, as a drawback, the solution can also include parameters that are not really realistic. This is hard to judge in the current state, since some parameters have been varied significantly and some additional description on all processes might be needed.
For example, there might be some issues with which the changes in "Southern Ocean wind stress" are described. I think the authors start the LGM with weaker southern hemispheric westerlies, and these are linearly increased over the deglaciation. While Table 2 represents changes in parameter values across the deglaciation (e.g. a land carbon uptake of 445, 890 or 1335 GtC), the Southern Ocean windstress is marked as decreasing. This is also the case in Figure 7, with the notation (-10 to -60%), whereas, clearly the impact of increased southern hemispheric westerlies is shown. Same in the notation of Figure 9.
P23, paragraph starting L.8: It might be easier to follow if everything goes in the sense of the deglaciation, i.e. describe an increase in the SO winds.

Similarly, due to changes in winds and sea-ice the Southern Ocean gas exchange should increase across the deglaciation.

A change in the remineralization profile across the deglaciation is fine, and remineralization would be expected to become shallower. However, it is not clear to me what is the rationale behind choosing "a linear profile". In between which depths was the profile made linear?

Based on the "best guess scenarios" of the parameters space, what is the most plausible change in land carbon?

3) I am a bit surprised by the sign of the oceanic d13C change when the SO wind increases. From figure 9, it seems like the stronger wind does not increase deep ocean ventilation: Tschumi et al., 2011 simulates positive d13C anomalies in the deep Pacific.

4) In some of the experiments described in the manuscript it seems that global alkalinity changes. For example, p. 19, L.10: "a deglacial decrease in deep Pacific carbonate ions" is needed to explain the deglacial CO2 increase. Since [CO3]~[ALk]-[DIC], and since deep Pacific DIC is expected to decrease across the deglaciation, that means that the alkalinity must decrease more than DIC. Additional information on the magnitude and processes leading to the alkalinity decrease might be needed.

Minor points and typos:
- P3, L.32: please add "across the deglaciation"
- P4, section 2.1: A minimum information about the Bern3D is needed without having to go through the appendix: which components does the model include? What is the resolution of the model?
- P9, L.18: Please spell out "std"
- P10, para L. 14: This paragraph might need a bit of rephrasing. I think the argument of the authors here is that changes in Fe fertilization lead to changes in the oceanic organic matter content. These changes can somewhat be broadly included in the changes associated with "organic weathering" in their set of simulations.
- P27, L. 5: a parenthesis is missing
- P27, L. 8: "phosphorus"
- P29, L.5: "biogeochemical"

---

## Referee Comment (RC2) · Katsumi Matsumoto (Referee) · 10 Feb 2019

This ms presents new modeling results of carbon redistribution in an earth system model, focusing on the change in land biosphere carbon storage between LGM and PI. It considers open system processes (i.e., weathering and burial) and uses observations of d13C of atm CO2, d13C of oceanic DIC, atm pCO2, and deep ocean [CO3=] as constraints. Since the seminal work by Shackleton (1977) the change in carbon storage on land over glacial-interglacial time scale has been an important topic in paleoclimatology and global carbon cycle. This ms is thus appropriate for CP readership.

The extensive simulations and analysis will add to the literature, and I am generally supportive of the study. However, I find the current form of ms requires some effort on the part of the reader to finish reading to the end, because it is quite long, a bit repetitive, and unclear in some places. Consequently this manuscript may not be as impactful as it can be. I make the following comments/questions/suggestions as a way to help improve this ms generally and increase its impact.

1) Overall the ms is long. There is some repetition of information in the Discussion that rehashes the Introduction without necessarily adding value. Effort should be made to shorten not just the Discussion but earlier sections too.

2) Even while the title and Figure 3 try to impress that reader that the authors are investigating the entire deglacial, I do not get a full sense of this. Most of the study is actually focused on differences between two times, LGM and PI (Figs 5-9). There is not a single figure that shows model results of the entire transient deglacial that corresponds to the deglacial forcing shown in Fig 3.

Related to this title-content mismatch is that the authors present Delta (e.g., Dland) as PI minus LGM. I find this to be unnecessarily confusing, because what is changing is the simulation results of LGM. PI remains the same. A main focus is on explaining LGM land carbon storage. In their way of presenting, when LGM land C increases, Dland decreases. As a reader, I have to do that mental calculation and commit what little RAM I have in my brain to storing that information...making reading difficult. For example, as an explanation of Fig 9, I found it a lot easier to understand "a weaker and shallower Atlantic Meridional Overturning Circulation at the LGM compared to the PI" (line 6, p 25) than "the deepening and strengthening of the AMOC during the deglaciation lead to positive Dd13C_DIC changes in the deep Atlantic..." (line 24-26, p 21). They are saying the same but PI-LGM is harder, at least for me.

My preference/suggestion therefore would be that the authors focus on time slice comparisons and present Delta as LGM-PI.

[Figure]

3) One of the main conclusions is that glacial land C was not larger than PI land C, thus refuting Zimov and Zech… This is a little bit of attacking the straw man, because the notion that glacial land C was larger is highly speculative to begin with, as the authors note (line 21-23, page 2). The more serious and "arguably most reliable" (line 33, page 2) hypothesis of glacial land C is that constrained by C isotopes by Shackleton. It seems to me that the main scientific contribution of this paper is that it credibly revised upwards Shackleton's estimate by considering open system processes, not refuting a straw man argument. I would suggest that the ms be rewritten to make clear what the real scientific contribution is.

4) Why is the control PI run biased high in d13C of DIC? (line 11, page 6)

5) I realize that details of ocean biogeochemistry is not central to this ms, but some more description would have been nice…e.g., how many phytoplankton functional types are there that are limited by P, Fe, and Si? (line 13, page 30)? How much did new production change in the "standard" glacial run compared to PI in terms of Dd13C (Fig 9a) or contribute to the DCO2=27.8 ppm? The authors adjusted the remin depth scale later to modify the impact of new production, but was it that the new production was not sensitive to nutrient supply changes in the standard run?

6) To me, the pulse experiments (section 3.1) are novel and the most interesting part of this study. It shows that neglecting the open system processes, as earlier studies have done, leads to underestimation of Dland. I think though, this section needs more clarification and discussion.

I was confused about the 100 Gt negative pulse emission in a closed system (line 26-29, page 12)…what this is supposed to represent, because in reality 100 Gt does not disappear. It gets redistributed. I think it's supposed to represent the terrestrial uptake, no? If so, it should be stated plainly. Otherwise, you'd have to put it into the ocean (after all, it's a closed system), so that d13C of atm CO2 should increase but d13C of DIC should decrease (not increase as in Fig 4B unless 100 Gt is magically removed or

taken up by land).

By the way, I found the inverted y-axis of d13Catm (Fig 4A) to be confusing. Why invert? The reader has to commit that to memory as well.

Why does d13C_atm recover faster than pCO2 (Fig 4A, line 4, page 13)? In standard chemical oceanography, we learn that C isotope equilibration is a lot longer than CO2 chemical equilibration in the surface ocean (roughly 10 yrs. vs. 1 yr).

What is the authors' definition of achieving equilibrium (line 11, page 13)? 60 kyr seems quite long as a time scale of carbonate compensation.

The most interesting part is the open system response of d13C of DIC (line 3-16, page 14). I feel that the main message is a little bit lost in the details. Is another way of putting it that open system damps out the penetration signal of light d13C from the atmosphere into the ocean? So when benthic foraminifera records seawater d13C, its perturbation signal would appear to be much smaller than closed system/actual, and therefore the reconstructed Dland would be underestimated?

I think you mean, compared to the "closed system," not the "open system" in line 14, page 13.

I don't understand "the removal of light terrestrial carbon" in line 5-6, page 14.

I think this whole section should be more carefully worded, perhaps even expanded, so that its important message is up front and clearer.

7) Pages 16-17 make clear that d13C of DIC provides the single most important con-straint on Dland, which I believe is what most people in the community thought anyway: "arguably most reliable" (line 33, page 2). So it seems to me that, in the bigger scheme of things, the place of this work in the literature is that it builds on the C isotope-estimated Dland by making the important point that open system processes lead to underestimation. I think that is how the ms should be framed, not refuting straw man argument of Zimov or trying to explain glacial pCO2. . .

As such the rest of the ms is interesting (especially Figs 7 and 8), but it enters the familiar territory of trying to explain DpCO2~90 while being consistent with observational constraint. . .concluding that more than one mechanism would be required to explain the full amplitude. I think David Archer was in fact the first to say that it takes multiple mechanisms. . .not the later papers cited in lines 27-28, page 27. There is very good discussion about the individual mechanisms but in the end, this ms does not point to a feasible combination of mechanisms that would solve the glacial CO2 problem satisfactorily. In fact, looking at Figs 7 and 8, I get the sense that these mechanisms in any combination cannot satisfy CO2 and C isotopes targets. For example, observations indicate a larger change in d13C of DIC than d13C of atm CO2, but none of the mechanisms have the right slope (in Fig 8D). Also, some of the findings seem quite obvious: e.g., that CO2 is inversely related to CO3= (lines 7-12, page 19; Fig 7C). . .that seems pretty standard chemical oceanography stuff.

So while Figs 7 and 8 are interesting and illuminating, I feel there is a lot of text devoted to a topic that is rather familiar without reaching a satisfactory solution. I think if the text after section 3.1 is shortened, the novelty of the ms becomes more apparent and ultimately increases its impact.

8) The ms has awkward phrasings/words here and there, which need to be rephrased. The following is not an exhaustive list:

- is "paleo soils" (line 22, p.2) correct? is it "paleosols"

-replace "by" with "to" (line 24, p. 3)

-insert "on land" after "organic material" (line 10, p.10)

-insert "the pulse to" after "apply" (line 27, p. 12)

-insert "marine" before "sediments" (line 10, p. 13)

-last line, p.13 is confusing; rephrase

-replace "recognize" with "represent" and rephrase rest of sentence (line 33, p. 14)

-line starting "Also, it is visible. . ." (line 7, p. 17) is confusing; rephrase

-line starting "In summary. . ." (line 28, p. 17) makes little sense; rephrase

-insert "of" at the end of line 16, p. 19 after "mechanism"

-line starting "Though, the overall. . ." (line 18, p. 19) is awkward; rephrase

-replace "The following" with "Subsequent" on line 23, p. 10

-replace "build" with "built", line 19, p. 24

-line 30, p. 25: Is "Information in" necessary? Or are the authors inferring from Baldini's study? Awkward in any case. . .rephrase.

-misspelling of phosphorus, line 8, p. 27

-a brief description of the 4 box veg model might be helpful

---

## Author Comment (AC1) · 14 Mar 2019

**Response to the review comments**

We thank both reviewers for their time and effort to assess our manuscript and for their helpful comments and advice. In the reply, the original review comment is given in small fonts, the reply in normal fonts and changed manuscript text in blue. Page and line numbers are all given for the originally submitted manuscript

A manuscript with changes highlighted is attached to this reply.

**Review 1**

Jeltsch-Thommes et al., provide a new estimate of changes in land carbon across the deglaciation. This new estimate is obtained by finding the best fit between paleoproxy constraints and results of factorial simulations with the Bern3D Earth system model coupled to Monte Carlo ensemble.

Their results suggest that there was less land carbon at the LGM than during the Holocene. Their new estimate of deglacial change in land carbon is higher than inferred by recent studies due to the inclusion of sedimentary processes. I recommend publication in Climate of the Past, granted the comments below are taken into account.

Thank you for your support and thoughtful comments.

1) The estimate of land carbon change across the deglaciation given here is higher than previous ones due to sedimentary processes. Since land carbon and organic matter have similar d13C signatures, a deglacial increase in land carbon can be partly compensated by a decrease in organic carbon burial. It is true that in a pure mass balance approach these sedimentary might not be taken into account (even if after ~10k, these effects are not expressed fully, cf. Fig. 4). However, the model used here (Bern3D) includes a sediment model. Therefore changes in temperature, circulation, remineralization profile, etc… lead to changes in organic and carbonate burial. Sedimentary processes are therefore already taken into account in this modelling framework. I am thus confused by the inclusion of an additional "organic weathering flux". I can indeed imagine that an input of depleted d13C as would happen with a 100% increase in organic matter flux through river could significantly impact oceanic d13C. Maybe the rationale and numbers associated with the organic weathering flux should be explained in more details.

Done. While Bern3D includes an ocean sediment module to simulate dynamically the burial of organic material, the corresponding weathering flux is prescribed to be constant. This is explained in the appendix (page 31, line 7ff), but was not explained in the main manuscript text.

We added the following explanations at P10, L11: The weathering flux to the ocean (Tab. 1) is kept constant in the standard setup of the Bern3D (see also appendix A1), but might have varied with changes in climate and $CO_2$. To account for this possibility, we varied the input of …

and added to the caption of Tab. 1, P7: The tracer input flux to the ocean resulting from weathering is set to compensate for the burial fluxes as diagnosed at the end of the spin up and kept constant in Bern3D standard simulations.

2) [2a] As a follow up on the previous comment, large parameters studies like this one are very useful to test the range of possibilities and derive statistics on possible parameter space. However, as a drawback, the solution can also include parameters that are not really realistic. This is hard to judge in the current state, since some parameters have been varied significantly and some additional description on all processes might be needed.

Figure C2 on P36 shows to which extent the parameters are varied a priori and in the constrained ensemble. We added further text to address this point.

On P9, L10 for the wind stress scaling: We do not consider such large changes in wind stress as realistic, but rather use prescribed Southern Ocean wind stress as a tuning knob to vary deep ocean ventilation in the Bern3D model. Consequently, we do not scale air-sea gas transfer rates with the changes in wind stress, but vary transfer rates independently.

and on P9, L27 for CaCO3 deposition: [Here, we scale the reconstructed deposition history based on Vecsei and Berger (2004) (Fig. 3D) by applying a constant scaling] to vary $CaCO_3$ deposition within the published range.

In addition, we added a reference to Figure C2 panel D on P27, L18 and on P9, L31 for the rain ratio we change the last sentence to now read: There is a lack of estimates how the rain ratio has varied in the past. Here, we vary the global rain ratio for LGM conditions (see also Appendix A; Fig 3F; Tab. 2) between 0.045 and 0.098, while the rain ratio is 0.083 in the standard setup and for PI conditions.

and modified the sentence on P27, L6 by adding a reference to Figure C2 and providing additional literature references for the organic matter remineralization depth: A modest, and plausible (Bendtsen et al., 2002;Matsumoto et al., 2007), deepening of the remineralization of organic matter in the upper ocean (see Fig. C2 panel E) leads to …

[2b] For example, there might be some issues with which the changes in "Southern Ocean wind stress" are described. I think the authors start the LGM with weaker southern hemispheric westerlies, and these are linearly increased over the deglaciation. While Table 2 represents changes in parameter values across the deglaciation (e.g. a land carbon uptake of 445, 890 or 1335 GtC), the Southern Ocean windstress is marked as decreasing. This is also the case in Figure 7, with the notation (-10 to -60%), whereas, clearly the impact of increased southern hemispheric westerlies is shown. Same in the notation of Figure 9.

Done. Changed labels in Figs. 7 and 9 to be consistent with Table 2 and added further explanation to the captions of Tab. 2 and Fig. 7 and 9.

Table 2: [Overview of mechanisms and parameters considered in factorial sensitivity experiments. Bold entries mark the standard parameter values.] Values represent  the export rain ratio at LGM, the change in remineralization profile (see text) at LGM, the scaling applied to the standard shallow water carbonate deposition history, and the land carbon uptake over the deglacial in GtC. In the case of SO wind stress, SO gas transfer rate, and the organic weathering flux, values represent the ratio of LGM to PI forcing.

Fig. 7: [Sensitivities are shown relative to the standard forcings (white dots).] Values in panel (C) correspond to the entries in Tab. 2.

Fig. 9: [Simulated deglacial change in the isotopic signature of DIC, $\Delta\delta^{13}C_{DIC}$ (PI minus LGM). Values are displayed along sections across the Atlantic, Southern Ocean, and Pacific and in response to standard deglacial forcings and (B) to (H) due to changes in individual mechanisms as inferred from factorial simulations.] $\Delta\delta^{13}C_{DIC}$ is shown for a deglacial carbon uptake of 445 GtC by the land biosphere (B), Southern Ocean wind stress (C) and gas transfer rate (D) scaled by a factor of 0.4 during the glacial relative to PI, rain ratio reduced to 0.045 during the glacial(E), deglacial shallow water carbonate deposition scaled by a factor of 4 (F), a change in the organic matter remineralization between a linear profile during the glacial and the standard depth-scaling (G), and for a twofold increase in the organic weathering flux during the glacial compared to PI (H). [The absolute change in….]

[2c] P23, paragraph starting L.8: It might be easier to follow if everything goes in the sense of the deglaciation, i.e. describe an increase in the SO winds.

We modified the paragraph as requested and in addition clarified the description of the experimental setup at various places in the manuscript.

First, we note that the experimental setup used in this study differs from a classical step-change approach as, e.g., in Tschumi et al. (2011). We clarified the forcing history in the method section by replacing the sentence on P7, L9-10 with: The forcing history for changes in Southern Ocean wind stress, Southern Ocean gas transfer rate, the rain ratio, the remineralization profile, and the organic weathering rate are prescribed following an idealized glacial-interglacial evolution. PI values are set to LGM values at 40 yrBP, kept constant from 40 kyrBP to 18 kyrBP, scaled back to the PI value over the termination (18 kyrBP to 11 kyrBP) and kept constant thereafter (Fig. 3F and Tab. 2).

Paragraph starting P23, L8 text now reads: Changes in the other four mechanisms affect the spatial pattern of $\Delta\delta^{13}C_{DIC}$ significantly (Fig. 9C, D, E, and G). We note that these patterns result from adjustment processes acting on different timescales. We recall the forcing history for changes in Southern Ocean wind stress, Southern Ocean gas transfer rate, the rain ratio, the remineralization profile, and the organic weathering rate; PI values are set to LGM values at 40 kyrBP, kept constant from 40 kyrBP to 18 kyrBP, scaled back to the PI value over the termination (18 kyrBP to 11 kyrBP) and kept constant thereafter (see also Fig. 3 and Table 2). Results are therefore not directly comparable to the step-change experiments described by Tschumi et al. (2011). As discussed in detail by Tschumi et al. (2011) and Menviel et al. (2015), a more poorly ventilated ocean due to reduced wind stress leads to an increase in $\delta^{13}C_{atm}$ and a subtle initial decrease in $\delta^{13}C_{DIC}$. Changes in temperature and circulation in response to decreased Southern Ocean wind stress at the beginning of our simulation (Fig. 3F) lead to lower marine oxygen concentrations and thus an increase in POC sedimentation. Subsequent removal of $^{13}C$ depleted carbon from the ocean increases $\delta^{13}C_{DIC}$. Restoring wind stress to its initial value over the deglacial reverses the changes. As the sediment feedbacks act on longer timescales, propagation of the signal to the whole ocean is not yet achieved. A complex interplay of processes such as changes in circulation and subsequent changes in export fluxes, oxygen concentrations, and remineralization of POC as well as changes in the lysocline and $CaCO_3$ cycling spatially overlay each other, yielding the $\Delta\delta^{13}C_{DIC}$ distribution seen in Fig. 9C.

Further, Fig. 3 was changed to show the whole setup, not only the deglacial and the following two sentences added to the figure caption: The freshwater forcing in the North Atlantic is varied over the last 40 kyr following Menviel et al. (2012). Dashed green lines indicate spin-up values.

[2d] Similarly, due to changes in winds and sea-ice the Southern Ocean gas exchange should increase across the deglaciation.

See comment to previous point. Text on P23, L19 was changed to read: [Decreasing the gas transfer rate] at the beginning of the 40 kyr simulation (Fig. 3F) [leads to a strong increase in...]

A change in the remineralization profile across the deglaciation is fine, and remineralization would be expected to become shallower. However, it is not clear to me what is the rationale behind choosing "a linear profile". In between which depths was the profile made linear?

For clarification we show the linear profile here (Figure 1) and added the following text on P10, L7: [...] and a linear profile (75 m to 2000 m).

[Figure]

*Figure 1: Fraction of remaining particulate organic matter (indicated by the amount of particulate organic phosphorus, POP) leaving the euphotic zone for the standard and linear remineralization profile. $z_0$ refers to the bottom of the euphotic zone in the model at 75 m.*

and moved and streamlined the text from P27, L11-17 to P10, L7: [The modification in the remineralization profile of particulate organic matter is different than those applied by Menviel et al. (2012) and Roth et al. (2014) in simulations with the Bern3D model. Here, only the remineralization profile in the upper 2,000 m is changed and, in contrast to these earlier studies, it is assumed that the fraction of exported particles that reach the sediments below 2,000 m is not altered. The implicit assumption is that cooler temperatures at the LGM only significantly affect the dissolution of relatively small or labile particles in the upper ocean, while large or refractory particles sink fast enough to reach ocean sediments both under LGM and PI conditions].

Based on the "best guess scenarios" of the parameters space, what is the most plausible change in land carbon?

We are not sure we understand the question of the reviewer. Change in land carbon storage is determined in the Bayesian framework and results in a median value of 850 GtC as given in the abstract. Change in land carbon is used as one of seven input parameters for the emulator and as such difficult to assess as a function of the other six forcing parameters.

3) I am a bit surprised by the sign of the oceanic d13C change when the SO wind increases. From figure 9, it seems like the stronger wind does not increase deep ocean ventilation: Tschumi et al., 2011 simulates positive d13C anomalies in the deep Pacific.

Several aspects lead to the differences in oceanic $\delta^{13}C$ change as shown in Fig. 9C versus those given by Tschumi et al. in their Fig. 9. Most important is the different setup of the experiments as discussed above. When applying a step-increase in Southern Ocean wind stress to 180% as done in Tschumi et al. (2011), we obtain very similar results as illustrated in Figure 2 here below. Further differences arise from the different model resolution used by Tschumi et al. (2011) and in this study. See our answer to point [2c] above for changes to the MS.

[Figure]

*Figure 2: Atlantic and Pacific zonal means of $\delta^{13}C_{DIC}$ perturbation after 15 kyr in response to an 80% increase in Southern Ocean wind stress.*

4) In some of the experiments described in the manuscript it seems that global alkalinity changes. For example, p. 19, L.10: "a deglacial decrease in deep Pacific carbonate ions" is needed to explain the deglacial CO2 increase. Since [CO3]~[ALk]-[DIC], and since deep Pacific DIC is expected to decrease across the deglaciation, that means that the alkalinity must decrease more than DIC. Additional information on the magnitude and processes leading to the alkalinity decrease might be needed.

We added the following sentences on P19, L10 to provide an insight into DIC and Alk changes: Changes in alkalinity and DIC (PI-LGM) for the standard forcing amount to about 500 Gt eq and 365 GtC, respectively. In the case of rain ratio, coral reef growth, and changes in the remineralization profile, changes in alkalinity and DIC are largest and occur close to a 2:1 ratio. Changes in land carbon uptake

remove carbon from the system, leading to changes in alkalinity via $CaCO_3$ compensation whereas in the case of the organic weathering flux, not only carbon is added/removed from the system but also nutrients and alkalinity, following Redfield ratios (see appendix \ref{app:b3d}). Changes in alkalinity and DIC resulting from changes in Southern Ocean wind stress and gas transfer rate are small.

Minor points and typos:

- P3, L.32: please add "across the deglaciation"

Done

- P4, section 2.1: A minimum information about the Bern3D is needed without having to go through the appendix: which components does the model include? What is the resolution of the model?

The following text is added at P4, L25: It couples a dynamic geostrophic-frictional balance ocean, a thermodynamic sea ice component, and a single layer energy-moisture balance atmosphere. The horizontal resolution is 41x40 grid cells and the ocean has 32 layers. Marine productivity is simulated as a function of nutrient concentrations (P, Fe, Si), temperature and light and transferred to dissolved organic and particulate matter. Biogenic matter decomposes within the water column. Opal, calcite, and organic particles reaching the ocean floor are entering reactive sediments. A 10-layer sediment model is used to compute fluxes of carbon, nutrients, alkalinity and isotopes between the ocean, reactive sediments, and the lithosphere. Loss fluxes to the lithosphere are compensated at equilibrium by a corresponding input flux from weathering. A 4-box reservoir model is used to calculate the dilution of atmospheric isotopic perturbations by exchange with the land biosphere.

- P9, L.18: Please spell out "std"

Done

- P10, para L. 14: This paragraph might need a bit of rephrasing. I think the argument of the authors here is that changes in Fe fertilization lead to changes in the oceanic organic matter content. These changes can somewhat be broadly included in the changes associated with "organic weathering" in their set of simulations.

Done. We replaced the sentence on L15 ("The effect of iron fertilization on oceanic carbon and $\delta^{13}C_{DIC}$ is similar to the mechanisms outlined above and is thus not explicitly considered further.") by:

Iron fertilization leads to changes in the cycling of marine organic material and to related changes in the whole ocean inventory of $\delta^{13}C_{DIC}$ and other variables. These changes are broadly similar to the changes caused by altering the organic matter remineralization depth or the weathering input flux and implicitly included in the Monte Carlo simulations. Similar, other ..

- P27, L. 5: a parenthesis is missing

Done

- P27, L. 8: "phosphorus"

Done

- P29, L.5: "biogeochemical"

Done

**Review by Katsumi Matsumoto**

This ms presents new modeling results of carbon redistribution in an earth system model, focusing on the change in land biosphere carbon storage between LGM and PI. It considers open system processes (i.e., weathering and burial) and uses observations of d13C of atm CO2, d13C of oceanic DIC, atm pCO2, and deep ocean [CO3=] as constraints. Since the seminal work by Shackleton (1977) the change in carbon storage on land over glacial-interglacial time scale has been an important topic in paleoclimatology and global carbon cycle. This ms is thus appropriate for CP readership.

The extensive simulations and analysis will add to the literature, and I am generally supportive of the study. However, I find the current form of ms requires some effort on the part of the reader to finish reading to the end, because it is quite long, a bit repetitive, and unclear in some places. Consequently this manuscript may not be as impactful as it can be. I make the following comments/questions/suggestions as a way to help improve this ms generally and increase its impact.

Thank you for your support and for your useful comments.

1) Overall the ms is long. There is some repetition of information in the Discussion that rehashes the Introduction without necessarily adding value. Effort should be made to shorten not just the Discussion but earlier sections too.

We deleted/shortened/streamlined text in several places in the manuscript. Large changes are listed below, small changes are highlighted in the attached diff file for better readability.

- P1, L1-L6. First sentence of abstract deleted and following text shortened to read: Past changes in the inventory of carbon stored in vegetation and soils remain uncertain. Earlier studies inferred the increase in the land carbon inventory (Δland) from the Last Glacial Maximum (LGM) to the recent preindustrial period (PI) from reconstructions of the stable carbon isotope ratio in the ocean and atmosphere with recent estimates yielding 300-400 GtC. Surprisingly…

- P1, L14-18: Text modified and shortened to read: Our study demonstrates the importance of ocean-sediment interactions and burial and weathering fluxes involving marine organic matter to explain deglacial change and suggests a major upward revision of earlier isotope-based Δland estimates.

- P1, L12: new sentence added. It is highly unlikely that the land carbon inventory was larger at LGM than PI.

- P2, L18-32 on earlier estimates of changes in land carbon since LGM in the introduction; part of its content is included in section 4.3 in the discussion and deleted in the introduction.

- P3, L7: Sentence deleted ('The details of the approach...')

- P3 L14 -21

- P3, L29-31

- P4, L10-L12

- P5, we simplified Figure 1 and shortened the caption by deleting the text from L2 to L5 (The targets ..)

- P14, L33: Deleted sentence 'We now turn...'

- P16, L6-7: replaced sentence with: We stratify the results by Δland and define four sets...

- P17, L8-9: Deleted sentence 'Visual inspection...'

- P21, L27-28: Deleted sentence 'The small mean change...'

- P24, L9-10: Deleted sentence 'These geological...'

- Deleted paragraph starting P24, L19

- P24, L28-32: Shortened to read: There are a number of uncertainties in our approach. We rely on characteristic forcing–response relationship as obtained with the Bern3D model and these may be different than in reality. The current crop of models...

- P25, L25 to P26, L1: Shortened to read: The deglacial change in $CO_2$ stored in the unsaturated zone of aquifers is in the range of 12 to 86 GtC (Baldini et al., 2018) and small compared to the estimated change in land carbon of around 850 GtC. Perhaps more important, the modern inventory of refractory DOC is about 650 GtC (Hansell, 2013) and assumed constant in Bern3D, while changes in the small inventory of labile DOC are explicitly simulated. It is unclear how the DOC pool varied over the past. Further, [Cartapanis et al. (2016) suggest an approximately...]

- P26, L4-7: Shortened to read: Here, we varied organic-like carbon input by weathering in the unconstrained ensemble over a range corresponding to a cumulative PI-LGM anomaly of +780 to -1,560 GtC. [These anomalies in input...]

- P26, L12-19: Deleted

- P26, L20-35: Shortened to read: Our approach is limited in that we only consider the change between LGM and PI for four relevant proxy variables. This restriction allowed us to build a cost-efficient and non-linear emulator to explore a very large range of parameter combinations. Future efforts to refine estimates of Δland may explicitly consider the spatial and temporal evolution for a large number of proxies.

- P27, L18-21: Deleted ('A decrease in…' until '...by organic material.'

- P27, L33-P28, L1 deleted 'each associated with its own uncertainties and limitations.'

- P27, L25-30: Deleted here and last two sentences moved to beginning of section 4.

- P28, L3-5: Deleted sentence and shortened.

- P28, L8-15: Shortened to read: The ice core $CO_2$ and $\delta^{13}C_{atm}$ records constrain the preindustrial terrestrial carbon uptake over the Holocene to about 250 GtC (Elsig et al., 2009). Assuming the total estimate of 1,500 GtC by Adams and Faure (1998) was right...

- P28, L30-31: Deleted sentence

2) [2a] Even while the title and Figure 3 try to impress that reader that the authors are investigating the entire deglacial, I do not get a full sense of this. Most of the study is actually focused on differences between two times, LGM and PI (Figs 5-9). There is not a single figure that shows model results of the entire transient deglacial that corresponds to the deglacial forcing shown in Fig 3.

[2b] Related to this title-content mismatch is that the authors present Delta (e.g., Dland) as PI minus LGM. I find this to be unnecessarily confusing, because what is changing is the simulation results of LGM. PI remains the same. A main focus is on explaining LGM land carbon storage. In their way of presenting, when LGM land C increases, Dland decreases. As a reader, I have to do that mental calculation and commit what little RAM I have in my brain to storing that information. . .making reading difficult. For example, as an explanation of Fig 9, I found it a lot easier to understand "a weaker and shallower Atlantic Meridional Overturning Circulation at the LGM compared to the PI" (line 6, p 25) than "the deepening and strengthening of the AMOC during the deglaciation lead to positive Dd13C_DIC changes in the deep Atlantic. . ." (line 24-26, p 21). They are  saying the same but PI-LGM is harder, at least for me.

[2c] My preference/suggestion therefore would be that the authors focus on time slice comparisons and present Delta as LGM-PI.

[2a] We changed the title to read: Low terrestrial carbon storage at the Last Glacial Maximum: constraints from multi-proxy data

[2a] Our goal is to answer a scientific question. Namely, how large is the difference in global land carbon storage between the LGM and the PI. This is spelled out in the abstract and the introduction and

conclusions. To answer this question, we carried out more than 100 transient simulations over the deglacial period and figure 3 illustrates the experimental setup and how the model is forced. We keep figure 3 as it is an integral and important part of our method description. We have the impression from this comment and also from comment 7 below that we failed to explain the main purpose of our deglacial simulations – this is to quantify the influence of deglacial processes on the whole ocean $\delta^{13}$C budget and therefore on estimates of Δland.

We modified the text in the introduction on P3, L29, P3, L32-34, on P4, L12, and reorganized the paragraph on P4, L13-L22 to read:

P3, L29: added text: These mechanisms should not be neglected when addressing the whole ocean budgets of carbon and $\delta^{13}$C to infer changes in land carbon stocks.

P3, L32-34, reorganized and expanded sentence: The transfer of $^{13}$C and carbon between the ocean-atmosphere system and reactive sediments and the lithosphere is affected by the well-documented reorganization of the marine carbon cycle across the deglaciation (e.g. Sigman and Boyle, 2000, Sigman et al., 2010; Fischer et al., 2010; Elderfield et al., 2012; Ciais et al., 2013; Martinez-Garcia et al., 2014; Jaccard et al., 2016; Cartapanis et al., 2016).

P4, L12-14 added text: As a consequence, these deglacial changes and their spatio-temporal evolution affect the mean $\delta^{13}$C signature of the global ocean in addition to changes in land carbon.

P4, L13-22; reorganized/modified paragraph: The primary goal of this study is to provide a new, observationally constrained best estimate of Δland recognizing the role of marine deglacial processes, weathering and burial, and sediment interactions in an Earth system model. To this end, we establish the response sensitivities for mean ocean $\delta^{13}$C and for other proxy targets (PI-LGM differences) to the changes in individual key deglacial carbon cycle mechanisms from transient simulations. An emulator of the Bern3D is built and the strengths of individual marine and terrestrial processes are varied in a large number of combinations to find all possible solutions for Δland that match a set of observational targets in a Bayesian Monte Carlo data assimilation framework (see Fig. 1). We further scrutinize the closed system assumption in the $^{13}$C mass balance approach with land carbon uptake as the only forcing. Specifically, we run idealized pulse-like carbon uptake simulations with the Bern3D Earth System Model of Intermediate Complexity (EMIC) both with and without simulating sediment interactions, and with and without interactive weathering.

We further discuss multi-proxy response relationships for different carbon cycle processes and briefly address the possible contribution of individual mechanisms to the deglacial change in atmospheric $CO_2$ and $\delta^{13}$C($CO_2$) and of deep ocean carbonate ion concentration. Our results demonstrate the importance of different deglacial mechanisms, of ocean-sediment interactions, and of the burial-weathering cycle for the whole ocean $\delta^{13}$C signature. All previous $\delta^{13}$C-based estimates of Δland neglected the influence of these processes and appear systematically biased low.

In agreement with the advice given by the reviewer in [2c], we continue to focus on the comparison between the LGM and PI time slice.

[2b] Please see also our answer to reviewer 1, comment [2c] and to your comment 1 and our revision of the abstract.

We appreciate the advice regarding the presentation of results and attempted to harmonize text formulations (as well as figures and tables) to the extent possible. Reviewer 1 and this reviewer, however, provide conflicting advice how to present the results. Reviewer 1 suggests (comment 2c) that "It might be easier to follow if everything goes in the sense of the deglaciation." We prefer to follow the advice of reviewer 1 as it is likely easier for most readers to go forward in time in the sense of the deglaciation. It might also be confusing to talk about a negative glacial-interglacial change in atmospheric $CO_2$ with $\Delta CO_2$= -90 ppm. Therefore, we continue to present changes as differences between PI minus LGM.

We disagree somewhat with the argument of the reviewer that "what is changing is the simulation results of LGM. PI remains the same." We are only able to constrain the *change* in land carbon storage, Δland , but do not attempt to estimate the PI carbon inventory. Further, we apply a transient protocol for our simulations which covers the past 40 kyrs as now clarified in the revised Fig. 3.

3) One of the main conclusions is that glacial land C was not larger than PI land C, thus refuting Zimov and Zech... This is a little bit of attacking the straw man, because the notion that glacial land C was larger is highly speculative to begin with, as the authors note (line 21-23, page 2). The more serious and "arguably most reliable" (line 33, page 2) hypothesis of glacial land C is that constrained by C isotopes by Shackleton. It seems to me that the main scientific contribution of this paper is that it credibly revised upwards Shackleton's estimate by considering open system processes, not refuting a straw man argument. I would suggest that the ms be rewritten to make clear what the real scientific contribution is.

Please see our revision of the abstract in response to your comment 1. For example, we do not cite the literature range for Δland of -400 to 1500 GtC range anymore in the abstract and emphasize the upward revision of earlier $\delta^{13}$C based estimates in the newly added last sentence.

We shortened the text on the papers by Zimov et al. and Zech et al. in the intro on P2, L21ff (see response to comment 1 above).

We state now at the end of the introduction (see also response to [2a] above): All previous $\delta^{13}$C-based estimates of Δland neglected the influence of these processes and appear systematically biased towards lower estimates.

A recent publication by Kemppinen et al., 2018 suggests based on a large ensemble of EMIC simulations that land carbon storage was larger at the LGM than PI. Thus, there are at least three studies that suggest a higher carbon storage at LGM than at PI and it seems not justified to entirely ignore these. We state in the revised abstract that a larger land carbon inventory at LGM than PI is highly unlikely.

4) Why is the control PI run biased high in d13C of DIC? (line 11, page 6)

Several reasons potentially lead to this constant offset. During the spin-up, the model was forced with a prescribed $\delta^{13}$C = -6.305‰, while $\delta^{13}$C = -6.4‰ might have been preferable for AD 1765 conditions. The implementation of fractionation during air-sea gas exchange and primary production in the ocean follows the literature (Mook, 1986; Freeman and Hayes, 1992). Already small uncertainties in these formulations will lead to offsets in the equilibrium values.

5) I realize that details of ocean biogeochemistry is not central to this ms, but some more description would have been nice. . .e.g., how many phytoplankton functional types are there that are limited by P, Fe, and Si? (line 13, page 30)? How much did new production change in the "standard" glacial run compared to PI in terms of Dd13C (Fig 9a) or contribute to the DCO2=27.8 ppm? The authors adjusted the remin depth scale later to modify the impact of new production, but was it that the new production was not sensitive to nutrient supply changes in the standard run?

P30, L15: Text added: Bacteria and plankton are not explicitly represented.

P17, L16: Text added: New production of organic matter decreases from 11.98 to a mean value over the glacial of 11.25 GtC/yr.

We did not quantify the influence of the new production change on atmospheric $CO_2$. This is beyond the scope of this study. This would require additional factorial simulations and an adjusted model setup. The quantification is also not straightforward as ocean circulation changes affect the three classical carbon pumps all simultaneously.

6) To me, the pulse experiments (section 3.1) are novel and the most interesting part of this study. It shows that neglecting the open system processes, as earlier studies have done, leads to underestimation of Dland. I think though, this section needs more clarification and discussion.

[6a] I was confused about the 100 Gt negative pulse emission in a closed system (line 26- 29, page 12). . .what this is supposed to represent, because in reality 100 Gt does not disappear. It gets redistributed. I think it's supposed to represent the terrestrial uptake, no? If so, it should be stated plainly. Otherwise, you'd have to put it into the ocean (after all, it's a closed system), so that d13C of atm CO2 should increase but d13C of DIC should decrease (not increase as in Fig 4B unless 100 Gt is magically removed or taken up by land).

The starting sentence of section 3.1, P12, L26 is modified to: We start by investigating how atmospheric $CO_2$ and $\delta^{13}C$ signatures evolve in response to carbon uptake by the land biosphere when all other forcings remain unchanged. To this end, 100 GtC of light carbon ($\delta^{13}C$=-24 ‰) are instantaneously (in the first time step of the year) removed from the model atmosphere to test the Bern3D response for different model setups (see Appendix A3). This yields the Green's functions (or impulse response functions) for carbon and $^{13}C$ shown in Fig. 4.

We added references for the $CO_2$  closed system pulse response on P13, L1 to point the reader to publications discussing the value of and the theory behind impulse response functions:  ..by a typical impulse-response recovery (e.g. Maier-Reimer and Hasselmann, 1987;Siegenthaler and Joos, 1992;Joos et al., 2013)

and the $\delta^{13}C$ response in the closed system P13, L5 ..(Fig 4A; Joos et al., 1996)

and the $CO_2$ open system response on P13, L10: The evolution of atmospheric $CO_2$ (see Archer et al., 1998) … , while we are not aware of a peer-reviewed  publication showing the isotopic response for the open system.

[6b] By the way, I found the inverted y-axis of d13Catm (Fig 4A) to be confusing. Why invert? The reader has to commit that to memory as well.

Axis adjusted as requested.

[6c]Why does d13C_atm recover faster than pCO2 (Fig 4A, line 4, page 13)? In standard chemical oceanography, we learn that C isotope equilibration is a lot longer than CO2 chemical equilibration in the surface ocean (roughly 10 yrs. vs. 1 yr).

Indeed, this is standard chemical oceanography knowledge, though maybe it is not understood. We added the following sentence on P13, L6: The difference in the response of $CO_2$ and $\delta^{13}C$ is related to their different properties. $CO_2$ represents a concentration, whereas $\delta^{13}C$ stands for the $^{13}C$ to $^{12}C$ isotopic ratio. The capacity of the ocean to remove an atmospheric perturbation in $CO_2$, and equally in $^{12}CO_2$ and $^{13}CO_2$, is limited  by the acid-base carbonate chemistry described by (Revelle and Suess, 1957). In contrast, the removal of a perturbation in the isotopic ratio $^{13}CO_2/^{12}CO_2$ is hardly affected by this chemical buffering; the buffering affects $^{13}CO_2$ in the nominator and $^{12}CO_2$ in the denominator about equally leading to a near cancellation of its effect on the ratio. The resulting change...

Regarding the specific point about surface ocean equilibration by air-sea exchange, it may be useful to consider the atmosphere and surface ocean while not considering exchange with the deep. At equilibrium, the perturbation in the isotopic ratio for atmospheric $CO_2$ and surface ocean DIC become approximately equal. The atmosphere, in the form of $CO_2$, and the surface ocean, in the form of DIC, contain about equal mass of carbon. Accordingly, half of the initial perturbation in the isotopic ratio is removed from the atmosphere to the surface ocean.

In contrast, the acid-base carbonate chemistry leads to a much smaller relative perturbation in DIC than in $CO_2$ (acid-base buffering). With a Revelle factor of 10, roughly 90% of the initial perturbation remains in the atmosphere and only about 10% enters the surface ocean.

The surface ocean DIC concentration is equilibrated quickly as the surface layer only absorbs 10% of the initial perturbation.  In contrast, the isotopic equilibration is slower as a larger fraction of the perturbation has to be taken up. This leads to the faster decline in the impulse response function (IRF) of $\delta^{13}C_{atm}$

compared to the IRF of $CO_2$. This reasoning can be extended considering surface-to-deep mixing and the whole ocean DIC content.

[6d] What is the authors' definition of achieving equilibrium (line 11, page 13)? 60 kyr seems quite long as a time scale of carbonate compensation.

Text on P13,L11 modified to read: ... and equilibrium is approached with an e-folding time scale of about 14,000 years.

[6e]The most interesting part is the open system response of d13C of DIC (line 3-16, page 14). I feel that the main message is a little bit lost in the details. Is another way of putting it that open system damps out the penetration signal of light d13C from the atmosphere into the ocean? So when benthic foraminifera records seawater d13C, its perturbation signal would appear to be much smaller than closed system/actual, and therefore the reconstructed Dland would be underestimated?

Text from P13, L13 ff reformulated to read: [… thereafter.] The novel and most important finding from the pulse experiment is that the $\delta^{13}C$ perturbation in the ocean-atmosphere-land biosphere system is increasingly and eventually completely removed by open system processes. Fig. 4B shows the $\delta^{13}C_{DIC}$ perturbation to decrease on a multi-millennial timescale in contrast to the closed system assumption (dashed lines 4B). In other words, benthic foraminifera record a much smaller perturbation in seawater $\delta^{13}C$than expected from closed system modelling. The resulting difference in the $\delta^{13}C_{DIC}$ perturbation between the open and closed system causes an erroneous mass balance inference for the terrestrial biosphere, when applying conventional closed system equations (Eq. A2 and A3). On a timescale of 10 kyr, underestimation of the terrestrial carbon inventory change amounts to 30 %. The error increases as times evolves.

In detail, the difference between the open and closed system pulse responses is explained as follows. The $^{13}C$ budget in the open system changes due to imbalances between input and burial fluxes of POC and $CaCO_3$ and due to changes in the $\delta^{13}C$ signatures of the respective fluxes.

The uptake of isotopically light land carbon from the atmosphere leads to an increase in the $\delta^{13}C$ signature of surface waters and by that in the $\delta^{13}C$ signatures of exported and eventually buried POC and $CaCO_3$. The mean $\delta^{13}C$ signature of the burial flux amounts to -8.7‰ ($\delta^{13}C_{CaCO3}$: 2.9‰) and $C_{org}$ ($\delta^{13}C_{POC}$: -20.2‰) for the first 10 kyr compared to -9.1 ‰ for the weathering input. This difference tends to mitigate the $\delta^{13}C_{DIC}$ perturbation. Further, carbon–climate feedbacks cause changes in temperature and ocean circulation, which affect the export of POC and $CaCO_3$. Burial fluxes of POC further depend on $O_2$ concentrations and burial fluxes of $CaCO_3$ on $CO_3^{-2}$ concentrations which evolve over the course of the simulation. As a result, the cumulative sedimentation-weathering imbalance amounts to a removal of 48.5 GtC during the first 10 kyr. The contribution to differences in the carbon isotopic budget relative to the closed system is dominated by changes in POC cycling and its associated $\delta^{13}C$ signature, while changes in the $CaCO_3$ cycle play a smaller, but non-negligible, role. Overall, $^{13}C$ is lost from the atmosphere–ocean system and $\delta^{13}C_{DIC}$ is decreasing.

[6f] I think you mean, compared to the "closed system," not the "open system" in line 14, page 13.

Done, text deleted.

[6g]I don't understand "the removal of light terrestrial carbon" in line 5-6, page 14. I think this whole section should be more carefully worded, perhaps even expanded, so that its important message is up front and clearer.

Done, see response to 6e

7) [7a] Pages 16-17 make clear that d13C of DIC provides the single most important constraint on Dland, which I believe is what most people in the community thought anyway: "arguably most reliable" (line 33, page 2). So it seems to me that, in the bigger scheme of things, the place of this work in the literature is that it builds on the C isotope estimated Dland by making the important point that open system processes lead to underestimation. I think that is how the ms should be framed, not refuting straw man argument of Zimov or trying to explain glacial pCO2. . .

The statement of the reviewer on the importance of $\delta^{13}C_{DIC}$ is incorrect and not supported by our results – see Figure 5. We started a new paragraph on P15, L5 to make the point more readily accessible by the reader and modified the text from L5-8 to read:

Next, we show that it is not sufficient to use $\Delta\delta^{13}C_{DIC}$, or any other target, in isolation to constrain $\Delta$land. The probability distribution of $\Delta$land covers the whole range of sampled values almost uniformly (Fig. 5) when applying only one out of the four targets. Considering either $\Delta\delta^{13}C_{DIC}$ or $\Delta\delta^{13}C_{atm}$ together with $\Delta CO_2$ as constraints moves the $\Delta$land distribution towards the multi proxy-based distribution (blue line in Fig. 5; not shown). We conclude that the incorporation of multiple constraints is essential to narrow uncertainties in $\Delta$land.

Please see our response to your comments 1 and 3 with respect to the manuscript framing and Zimov et al.

[7b] As such the rest of the ms is interesting (especially Figs 7 and 8), but it enters the familiar territory of trying to explain DpCO2_90 while being consistent with observational constraint… concluding that more than one mechanism would be required to explain the full amplitude.

[7c] I think David Archer was in fact the first to say that it takes multiple mechanisms… not the later papers cited in lines 27-28, page 27.

[7d] There is very good discussion about the individual mechanisms but in the end, this ms does not point to a feasible combination of mechanisms that would solve the glacial CO2 problem satisfactorily. In fact, looking at Figs 7 and 8, I get the sense that these mechanisms in any combination cannot satisfy CO2 and C isotopes targets. For example, observations indicate a larger change in d13C of DIC than d13C of atm CO2, but none of the mechanisms have the right slope (in Fig 8D).

[7e]Also, some of the findings seem quite obvious: e.g., that CO2 is inversely related to CO3= (lines 7-12, page 19; Fig 7C)… that seems pretty standard chemical oceanography stuff. So while Figs 7 and 8 are interesting and illuminating, I feel there is a lot of text devoted to a topic that is rather familiar without reaching a satisfactory solution. I think if the text after section 3.1 is shortened, the novelty of the ms becomes more apparent and ultimately increases its impact.

[7b] We do not agree with the reviewer. Please see our response to your comment 2b above. In addition, we modified the last paragraph of section 3.1 to read:

The pulse responses make it clear that open system processes such as sediment interactions, burial, and weathering affect the carbon and $\delta^{13}C$ budgets on deglacial time scales. Taken at face values, the results suggest that the classical $\delta^{13}C$-based estimates of $\Delta$land are substantially biased low, because of the neglect of open system processes. However, uptake by land carbon is the only forcing in these idealized pulse simulations, whereas climate and biogeochemical cycles underwent a massive reorganization over the glacial-interglacial transition. This reorganization affected open system processes and the $^{13}C$ budget in addition to carbon uptake by the land biosphere. In the next section, we estimate $\Delta$land by taking into account deglacial processes.

[7c] Reference added.

[7d] The statement by the reviewer is incorrect. It is stated at several places in the MS that we use all four target variables as constraints and that all four variables must fall within the proxy range given in Tab. 2. Fig. 6 shows the kernel density estimates for the four targets in the constrained ensemble. In addition, Fig. C2 in the appendix gives the posteriori parameter distribution. We believe that this point is made sufficiently clear in the revised MS.

8) The ms has awkward phrasings/words here and there, which need to be rephrased. The following is not an exhaustive list:

- is "paleo soils" (line 22, p.2) correct? is it "paleosols"

Done

-replace "by" with "to" (line 24, p. 3)

Done

-insert "on land" after "organic material" (line 10, p.10)

Meant is the buried oceanic organic material.

-insert "the pulse to" after "apply" (line 27, p. 12)

Done

-insert "marine" before "sediments" (line 10, p. 13)

Done

-last line, p.13 is confusing; rephrase

Sentence changed

-replace "recognize" with "represent" and rephrase rest of sentence (line 33, p. 14)

Done

-line starting "Also, it is visible. . ." (line 7, p. 17) is confusing; rephrase

Done

-line starting "In summary. . ." (line 28, p. 17) makes little sense; rephrase

Shortened and changed

-insert "of" at the end of line 16, p. 19 after "mechanism"

Done

-line starting "Though, the overall. . ." (line 18, p. 19) is awkward; rephrase

Sentence deleted, information was not essential

-replace "The following" with "Subsequent" on line 23, p. 10

Done

-replace "build" with "built", line 19, p. 24

Done

-line 30, p. 25: Is "Information in" necessary? Or are the authors inferring from Baldini's study? Awkward in any case. . .rephrase.

Done

-misspelling of phosphorus, line 8, p. 27

Done

-a brief description of the 4 box veg model might be helpful

Provided in Appendix A

Archer, D., Kheshgi, H., and Maier-Reimer, E.: Dynamics of fossil fuel $CO_2$ neutralization by marine $CaCO_3$., Global Biogeochemical Cycles, 12, 259-276, 1998.

Bendtsen, J., Lundsgaard, C., Middelboe, M., and Archer, D.: Influence of bacterial uptake on deep-ocean dissolved organic carbon, Global Biogeochemical Cycles, 16, 74-71-74-12, 10.1029/2002GB001947, 2002.

Freeman, H. and Hayes, J. M.: Fractionation of Carbon Isotopes by Phytoplankton and Estimates of Ancient CO2 Levels, Global Biogeochemical Cycles, 6, 185–198, 1992.

Joos, F., Bruno, M., Fink, R., Siegenthaler, U., Stocker, T. F., and LeQuere, C.: An efficient and accurate representation of complex oceanic and biospheric models of anthropogenic carbon uptake, Tellus Series B-Chemical and Physical Meteorology, 48, 397-417, 1996.

Joos, F., Roth, R., Fuglestvedt, J. S., Peters, G. P., Enting, I. G., von Bloh, W., Brovkin, V., Burke, E. J., Eby, M., Edwards, N. R., Friedrich, T., Frölicher, T. L., Halloran, P. R., Holden, P. B., Jones, C., Kleinen, T., Mackenzie, F. T., Matsumoto, K., Meinshausen, M., Plattner, G.-K., Reisinger, A., Segschneider, J., Shaffer, G., Steinacher, M., Strassmann, K., Tanaka, K., Timmermann, A., and Weaver, A. J.: Carbon dioxide and climate impulse response functions for the computation of greenhouse gas metrics: a multi-model analysis, Atmos. Chem. Phys., 13, 2793-2825, 10.5194/acp-13-2793-2013, 2013.

Maier-Reimer, E., and Hasselmann, K.: Transport and storage of $CO_2$ in the ocean - an inorganic ocean-circulation carbon cycle model, Climate Dynamics, 2, 63-90, 1987.

Matsumoto, K., Hashioka, T., and Yamanaka, Y.: Effect of temperature-dependent organic carbon decay on atmospheric pCO2, Journal of Geophysical Research: Biogeosciences, 112, G02007, 10.1029/2006jg000187, 2007.

[revised manuscript text omitted]